# DECONFOUNDING REINFORCEMENT LEARNING IN OBSERVATIONAL SETTINGS

## ABSTRACT

In this paper, we propose a general formulation to cope with a family of reinforcement learning tasks in observational settings, that is, learning good policies solely from the historical data produced by real environments with confounders (i.e., the factors affecting both actions and rewards). Based on the proposed approach, we extend one representative of reinforcement learning algorithms: the Actor-Critic method, to its deconfounding variant, which is also straightforward to be applied to other algorithms. In addition, due to lack of datasets in this direction, a benchmark is developed for deconfounding reinforcement learning algorithms by revising OpenAI Gym and MNIST. We demonstrate that the proposed algorithms are superior to traditional reinforcement learning algorithms in confounded environments. To the best of our knowledge, this is the first time that confounders are taken into consideration for addressing full reinforcement learning problems.

## 1 INTRODUCTION

In recent years, reinforcement learning (RL) has made great progress, spawning a large number of successful applications especially in terms of games (Silver et al., 2016; Mnih et al., 2013; OpenAI, 2018). Within this background, much attention has been devoted to the development of RL algorithms with the goal of improving treatment policies in healthcare (Gottesman et al., 2018). In fact, various RL algorithms have been proposed to infer better decision-making strategies for mechanical ventilation (Prasad et al., 2017), sepsis management (Raghu et al., 2017a;b), and treatment of schizophrenia (Shortreed et al., 2011). In healthcare, a common practice is to focus on the observational setting, because ones do not wish to experiment with patients' lives without evidence that the proposed treatment strategy is better than the current practice (Gottesman et al., 2018) [1]. As pointed out in (Raghu et al., 2017a), even if in the observational setting, RL also has advantages over other machine learning algorithms especially in two situations: when the ground truth of a "good" treatment strategy is unclear in medical literature (Marik, 2015), and when training examples do not represent optimal behavior. On the other hand, although causal inference (Pearl, 2009) has been extensively explored and used in healthcare and medicine (Liu et al.; Soleimani et al., 2017; Schulam & Saria, 2017; Alaa et al., 2017; Alaa & van der Schaar, 2018; Atan et al., 2016), the efficient approach to dealing with time-varying data is still unclear (Peters et al., 2017; Hernn & Robins, 2018). On the basis of the discussion above, in this paper we attempt to combine advantages on both sides to cope with an important family of RL problems in the observational setting, that is, learning good policies solely from the historical data produced by real environments with *confounding bias*.

This type of problems are inevitable in the future RL research with the burgeoning development in healthcare and medicine. To the best of our knowledge, however, little work has been done in the direction. Confounding is a causal concept that is described in the language of causality instead of probability and statistics (Pearl, 2009). Confounding bias occurs when a variable influences both who is selected for the treatment and the outcome of the experiment (Pearl & Mackenzie, 2018), which naturally corresponds to the action and the reward in RL, respectively. As a matter of fact, confounders have been extensively studied in epidemiology, sociology, and economics. Take for example the widespread kidney stones in which the size of the kidney stone is a confounding factor affecting both the treatment and the recovery (Peters et al., 2017; Pearl, 2009), whether deconfounding the size of the kidney stone or not entirely determines how to choose a more effective treatment. Similarly, in RL, if unobserved potential confounders exist, they would affect both ac-

---

[1]Another straightforward example is that, in economics, considering the risky cost in terms of time and money, it is not practical to study the optimal strategy by actually buying and selling stocks in the market.

tions and rewards when an agent interacts with environments and eventually influence the policy to be optimized.

It is widely acknowledged that one should draw a causal graph before one can achieve any causal conclusion (Pearl, 2009; Pearl & Mackenzie, 2018). Throughout the paper, we assume that, given causal assumptions, we first estimate a model from the observational data we collected from real environments or simulators, and then optimize a policy on the basis of the learned model. As mentioned previously, this assumption is quite useful in real-world RL applications (e.g., healthcare, medicine, economics, etc), because in most circumstances, except the data we observed, we neither know anything about real environments nor are allowed to do anything in real environments probably for the sake of ethics, laws or cost.

In order to adjust for confounders, we present a general formulation for addressing this class of RL problems, namely *deconfounding reinforcement learning* (DRL). More specifically, given several common confounding assumptions, under some conditions for identification we first estimate a latent-variable model from observational data in which we simultaneously discover the latent confounders and infer how they affect action and reward, then deconfound the confounders using the causal language developed by (Pearl, 2009), and finally optimize the policy based on the deconfounding model we calculated. On the basis of the proposed formulation, we extend one popular RL algorithm, the Actor-Critic method, to its corresponding deconfounding variant, which is straightforward to be applied to other RL algorithms. Due to lack of datasets in this respect, we revise the classic control toolkit in OpenAI Gym (Brockman et al., 2016), making it a benchmark for comparison of DRL algorithms. In addition, we also devise a confounding version of the MNIST dataset (LeCun et al., 1998) to verify the performance of our causal model. Finally, we conduct extensive experiments to demonstrate the superiority of the proposed formulation in confounded environments, in comparison to traditional RL algorithms.

To sum up, our contributions in this paper are as follows:

1. We propose a general formulation to address a family of RL problems in confounded environments, namely *deconfounding reinforcement learning* (DRL);

2. We present the deconfounding variant of Actor-Critic methods, which is obvious to be applied to other RL methods;

3. We develop a benchmark for DRL by revising the toolkit for classic control in OpenAI Gym (Brockman et al., 2016) and by devising a confounding version of the MNIST dataset (LeCun et al., 1998);

4. We perform a comprehensive comparison of DRL algorithms with their vanilla versions, showing that the proposed approach has an advantage in confounded environments.

5. To the best of our knowledge, this is the first attempt to build a bridge between confounding and the full RL problem. This is one of few research papers aiming at understanding the connections between causal inference and the full RL.

## 2 BACKGROUND

In this section, we briefly review *confounding* in causal inference. We recommend Pearls excellent monograph for further reading (Pearl, 2009; Pearl & Mackenzie, 2018).

### 2.1 SIMPSON'S PARADOX

Let us begin with one of the most famous paradoxes in statistics: Simpson's Paradox. Consider the previously mentioned kidney stones, a classic example of Simpson's paradox (Peters et al., 2017). We collect electronic patient records to investigate the effectiveness of two treatments against kidney stones, where although the overall probability of recovery is higher for patients who took treatment $b$, treatment $a$ performs better than treatment $b$ on both patients with small kidney stones and with large kidney stones. More precisely, we have

$$p(R = 1|T = b) > p(R = 1|T = a); \qquad \text{but}$$
$$p(R = 1|T = b, Z = 0) < p(R = 1|T = a, Z = 0),$$
$$p(R = 1|T = b, Z = 1) < p(R = 1|T = a, Z = 1); \qquad (1)$$

where $Z$ is the size of the stone, $T$ the treatment, and $R$ the recovery (all binary). How do we cope with this inversion of conclusion? Which treatment do you prefer if you had kidney stones?

Does treatment $b$ cause recovery? The answers to these questions depend on the causal relationship between treatment, recovery, and the size of the kidney stone.

## 2.2 CONFOUNDING

An intuitive explanation for this kidney stone example of Simpson's paradox is that larger stones are more severe than small stones and are much more likely to be treated with treatment $a$, resulting in that treatment $a$ looks worse than treatment $b$. Therefore, it is straightforward to assume that the true underlying causal diagram of the kidney stone example is shown in Figure 1(a), where confounding occurs because the size of kidney stones influences both treatment and recovery. Here, the size of kidney stones is called confounder. The term "confounding" originally meant "mixing" in English, which describes that the true causal effect $T \rightarrow R$ is "mixed" with the spurious correlation between $T$ and $R$ induced by the fork $T \leftarrow Z \rightarrow R$ (Pearl & Mackenzie, 2018). In other words, we will not be able to disentangle the true effect of $T$ on $R$ from the spurious effect if we do not have data on $Z$. Conversely, if we have measurements of $Z$, it is easy to deconfound the true and spurious effects by adjusting for $Z$ that averages the effect of $T$ on $R$ in each subgroup of $Z$ (i.e., different size groups in the case of kidney stones).

## 2.3 DO-OPERATOR AND ADJUSTMENT CRITERION

From the viewpoint of causal inference, we can also use the language of intervention, namely *do-operator*, to formulate confounding. In fact, in the example of kidney stones, what we are interested in is how these two treatments compare when we force all patients to take treatment $a$ or $b$, rather than which treatment has a higher recovery rate given only the observational patient records. Mathematically, we focus on the true effect $p(R = 1|do(T = a))$ (i.e., *intervention distribution* where patients are forced to take treatment $a$) instead of the spurious effect $p(R = 1|T = a)$ (i.e., *observational distribution* where patients are observed to take treatment $a$). Therefore, as described previously, confounding can be naturally formulated by the discrepancy between $p(R|T)$ and $p(R|do(T))$.

Generally speaking, do-operator can be executed in two common ways: by Randomized Controlled Trials (RCTs) (Fisher, 1935) and by adjustment formulas (i.e., Back-door criterion and Front-door criterion) (Pearl, 2009). RCTs is the so-called golden standard but rather limited due to many impractical factors (e.g., safety, laws, ethics, physically infeasibility, etc.). Back-door and Front-door criterions require a known causal diagram in which causal assumptions are provided in advance. According to the Back-door criterion, in the kidney stone example, we can immediately attain

$$p(R = 1|do(T = a)) = \sum_{z=0}^{1} p(R = 1|T = a, Z = z)p(Z = z). \tag{2}$$

In fact, apart from the two adjustment formulas, we are provided with a syntactic method of deriving claims about interventions, namely *do*-calculus, developed by Pearl (Pearl, 2009). The *do*-calculus consists of three rules which can be repeatedly applied to simplify the expression for an interventional distribution. It offers a powerful tool to convert intervention probabilities to observation probabilities so that we can estimate the former from observational data alone.

## 2.4 PROXY VARIABLES FOR CONFOUNDING

If confounders can be measured, then they can be adjusted for through the methods we discussed in Section 2.3. However, in most cases where confounders are hidden or unmeasured, without further assumptions, it is impossible to estimate the effect of the intervention on the outcome. A common practice is to leverage observed proxy variables of the confounders (Angrist & Pischke, 2008; Maddala & Lahiri, 1992; Montgomery et al., 2000), which is one of the promising directions of exploiting big-data to conduct causal inference in the presence of multiple proxy variables for unmeasured confounders (Louizos et al., 2017). However, using proxy variables to correctly recover causal effects should meet strict mathematical assumptions (Louizos et al., 2017; Edwards et al., 2015; Kuroki & Pearl, 2014; Miao et al., 2018; Pearl, 2012; Wooldridge, 2009). Unfortunately, in practice, we do not know whether or not the complicated data from the real world meet those assumptions. Hence, following the same spirit from (Louizos et al., 2017), with multiple proxy variables available, under relaxed assumptions we estimate a latent-variable model in which we simultaneously discover the latent confounders and infer how they affect treatment and outcome.

## 3 DECONFOUNDING REINFORCEMENT LEARNING

In this section, we will formally introduce *deconfounding reinforcement learning* (DRL). Generally speaking, DRL consists of two steps: learning a deconfounding model shown in Figure 1(b) and optimizing a policy based on the learned deconfounding model. The main idea in step 1 is to simultaneously discover the hidden confounder and infer the causal effect through an estimation of

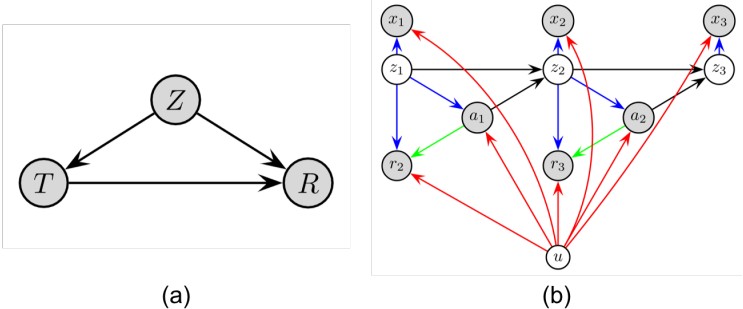

(a)                                                        (b)

Figure 1: (a) Causal diagram for kidney stones. (b) The model for deconfounding reinforcement learning. Solid nodes denote observed variables and open nodes represent unobserved variables. Red and blue arrows emphasize the observed variables affected by $u$ and by $z_t$, respectively. The causal effects of interest are colored in green.

a latent-variable model. More specifically, we first discuss the time-independent confounding assumption in Section 3.1, and then, based on the assumption, formalize the deconfounding model in Section 3.2. Section 3.3 talks about the problem of identification in our model, which is a central issue in causal inference. After that, we present the details about how to learn the model via variational inference in Section 3.4. Step 2 is provided in Section 3.5 where we describe how to design the deconfounding actor-critic method. It is straightforward to apply this to other RL algorithms in the same manner.

### 3.1 CAUSAL ASSUMPTIONS

Without loss of generality, as shown in Figure 1(b) [2], we assume there exists a common confounder in the sequential model, which is time-independent for each individual or for each procedure. This assumption is so general that it would apply to various RL tasks across domains. For example, in personalized medicine or precision medicine, socio-economic status can affect both the medication strategy a patient has access to, and the patients general health (Louizos et al., 2017). Therefore socio-economic status acts as confounder between the medication and health outcomes, in which case socio-economic status is time-independent for each patient during the course of treatment. In agriculture, soil fertility may serve as one of confounders affecting both the application of fertilizer and the yield of each plot (Pearl & Mackenzie, 2018). In this circumstance, soil fertility is stable and thought of as a time-independent factor within a period of time (e.g., several months, the growth circle of crops, etc.). In the example of stock markets, apart from socio-economic status as mentioned above, government policy may also act as one of confounders, all of which can be seen time-independent during a reasonable period of time.

### 3.2 THE MODEL

Given the causal assumption, we first fit a generative model to a sequence of observational data: observations, actions, and rewards, where actions and rewards are confounded by one or several unknown factors. Formally, Let $\vec{x} = (x_1, \ldots, x_T)$, $\vec{a} = (a_1, \ldots, a_{T-1})$, $\vec{r} = (r_2, \ldots, r_{T+1})$, $\vec{z} = (z_1, \ldots, z_T)$ be the sequence of observations, actions, rewards, and corresponding latent states, respectively. The confounder is denoted by $u$, and it is worth noting that here $u$ may stand for more than one confounder in which multiple confounders are seen as a whole represented by $u$. We assume that $x_t \in \mathbb{R}^{D_x}$, $a_t \in \mathbb{R}^{D_a}$, $r_t \in \mathbb{R}^{D_r}$, $z_t \in \mathbb{R}^{D_z}$, and $u \in \mathbb{R}^{D_u}$, where $D_z \ll D_x$. The generative model for DRL is then given by:

$$p(z_t) = \prod_{j=1}^{D_z} \mathcal{N}(z_{tj}|0, 1); \qquad p(u) = \prod_{j=1}^{D_u} \mathcal{N}(u_j|0, 1);$$

$$p(x_t|z_t, u) = \mathcal{N}\left(x_t|\hat{\mu}_t^x, \hat{\sigma}_t^{x\,2}\right); \quad \hat{\mu}_t^x = f_1(z_t, u) \quad \hat{\sigma}_t^{x\,2} = f_2(z_t, u); \qquad (3)$$

$$p(a_t|z_t, u) = \mathcal{N}\left(a_t|\hat{\mu}_t^a, \hat{\sigma}_t^{a\,2}\right); \quad \hat{\mu}_t^a = f_3(z_t, u) \quad \hat{\sigma}_t^{a\,2} = f_4(z_t, u); \qquad (4)$$

---

[2]Note that, in our case where a policy depending only on the confounder is applied to generating the data (Section 4.2), the arrow from $z_t$ to $a_t$ is not necessary when learning the model so that $z_t$ can be viewed as not a confounder of $a_t$ and $r_{t+1}$. This also provides another reason why we do not need to adjust for $z_t$. However, in some other cases such as when applied to medical data, the strategies of treatment from physicians definitely contain valuable information about $z_t$ and $a_t$, and therefore the arrow between them is necessary when learning the model.

$$p(r_{t+1}|z_t, a_t, u) = \mathcal{N}\left(r_{t+1}|\hat{\mu}_t^r, \hat{\sigma_t^r}^2\right); \quad \hat{\mu}_t^r = f_5(z_t, a_t, u) \quad \hat{\sigma_t^r}^2 = f_6(z_t, a_t, u); \quad (5)$$

$$p(z_t|z_{t-1}, a_{t-1}) = \mathcal{N}\left(z_t|\hat{\mu}_t^z, \hat{\sigma_t^z}^2\right); \quad \hat{\mu}_t^z = f_7(z_{t-1}, a_{t-1}) \quad \hat{\sigma_t^z}^2 = f_8(z_{t-1}, a_{t-1}). \quad (6)$$

Note that we parametrize each probability distribution as a Gaussian with its mean and variance modeled by nonlinear functions $f_k$ and each $f_k$ is parametrized by a neural network with its own parameters $\theta_k$ for $k = 1, \ldots, 8$. Note that Equation (4) is not necessary in our model, but it is potentially useful acting as a prior policy because it is learned from the observational data containing, for example, the real treatment strategies by doctors.

### 3.3 IDENTIFICATION OF CAUSAL EFFECT

In our deconfounding model shown in Figure 1(b), the key difference from traditional RL lies in the reward function [3]. To be more precise, assuming that an agent standing at state $z_t$ performs an action $a_t = \mathfrak{a}$, unlike the traditional reward $p(r_t|z_t, a_t = \mathfrak{a})$, our deconfounding version based on do-operator as depicted in Section 2.3 is given by

$$p(r_t|z_t, \text{do}(a_t = \mathfrak{a})) = \int_u p(r_t|z_t, \text{do}(a_t = \mathfrak{a}), u)p(u|z_t, \text{do}(a_t = \mathfrak{a}))\text{d}u, \quad (7)$$

$$= \int_u p(r_t|z_t, a_t = \mathfrak{a}, u)p(u)\text{d}u, \quad (8)$$

where Equation 8 is by the rules of *do*-calculus applied to the causal graph in Figure 1(b) (Pearl, 2009). In fact, we can utilize the back-door criterion to directly obtain Equation (8). Furthermore, through Equation (8) we proved that $p(r_t|z_t, \text{do}(a_t = \mathfrak{a}))$ can be identified from the joint distribution $p(u, \vec{z}, \vec{x}, \vec{a}, \vec{r})$. Then, motivated by the idea that one is supposed to use the knowledge inferred from the joint distribution between proxy variables and confounders to adjust for the hidden confounders (Louizos et al., 2017; Edwards et al., 2015; Kuroki & Pearl, 2014; Miao et al., 2018; Pearl, 2012; Wooldridge, 2009), in this paper we also assume that the joint distribution $p(u, \vec{z}, \vec{x}, \vec{a}, \vec{r})$ can be approximately recovered solely from the observations $(\vec{x}, \vec{a}, \vec{r})$. The details about identification can be found in Appendix A.

### 3.4 LEARNING

Since the nonlinear functions parametrized by neural networks make inference intractable, we will learn the parameters of the model $\theta_k$ by employing variational inference along with an inference model, a neural network which approximates the intractable posterior (Rezende et al., 2014; Kingma & Welling, 2013; Krishnan et al., 2015). More specifically, using the variational principle, we posit an approximate posterior distribution $q_\phi(z|x)$ to obtain the following lower bound on the marginal likelihood:

$$\log p_\theta(x) \geq \mathop{\mathbb{E}}_{q_\phi(z|x)} [\log p_\theta(x|z)] - \text{KL}\left(q_\phi(z|x)||p_\theta(z)\right), \quad (9)$$

where the inequality is by Jensen's inequality and $\phi$ is the parameter of the inference model $q(z|x)$. Note that in this general case $x$ stands for observational variables and $z$ for latent variables.

#### 3.4.1 VARIATIONAL LOWER BOUND

Directly applying the lower bound in Inequality (9) to our model, we obtain

$$\log p_\theta(\vec{x}, \vec{a}, \vec{r}) \geq \mathop{\mathbb{E}}_{q_\phi(\vec{z}, u|\vec{x}, \vec{a}, \vec{r})} [\log p_\theta(\vec{x}, \vec{a}, \vec{r}|\vec{z}, u)] - \text{KL}\left(q_\phi(\vec{z}, u|\vec{x}, \vec{a}, \vec{r})||p_\theta(\vec{z}, u)\right)$$

$$= \mathcal{L}(\vec{x}, \vec{a}, \vec{r}; \theta, \phi). \quad (10)$$

Using the Markov property of our model, the full distribution can be factorized in the following way:

$$p_\theta(\vec{x}, \vec{a}, \vec{r}, \vec{z}, u) = p(u)p(z_1)\left[\prod_{t=1}^T p(x_t|z_t, u)p(a_t|z_t, u)p(r_{t+1}|z_t, a_t, u)\right]\left[\prod_{t=2}^T p(z_t|z_{t-1}, a_{t-1})\right]. \quad (11)$$

In addition, for simplicity's sake, we also have the factorization assumption for the posterior approximation:

$$q_\phi(\vec{z}, u|\vec{x}, \vec{a}, \vec{r}) = q(u|\vec{x}, \vec{a}, \vec{r})q(z_1|\vec{x}, \vec{a}, \vec{r})\prod_{t=2}^T q(z_t|z_{t-1}, \vec{x}, \vec{a}, \vec{r}). \quad (12)$$

---

[3] An intuition can be found in Appendix F.2.

Combining Equation (10), (11) and (12) yields:

$$\log p_\theta(\vec{x}, \vec{a}, \vec{r}) \geq \mathcal{L}(\vec{x}, \vec{a}, \vec{r}; \theta, \phi)$$

$$= \sum_{t=1}^{T} \mathop{\mathbb{E}}_{\substack{z_t \sim q(z_t|z_{t-1}, \vec{x}, \vec{a}, \vec{r}) \\ u \sim q(u|\vec{x}, \vec{a}, \vec{r})}} [\log p(x_t|z_t, u) + \log p(a_t|z_t, u) + \log p(r_{t+1}|z_t, a_t, u)]$$

$$- \text{KL}\left(q(u|\vec{x}, \vec{a}, \vec{r})||p(u)\right) - \text{KL}\left(q(z_1|\vec{x}, \vec{a}, \vec{r})||p(z_1)\right)$$

$$- \sum_{t=2}^{T} \mathop{\mathbb{E}}_{\substack{z_{t-1} \sim \\ q(z_{t-1}|z_{t-2}, \vec{x}, \vec{a}, \vec{r})}} \left[\text{KL}\left(q(z_t|z_{t-1}, \vec{x}, \vec{a}, \vec{r})||p(z_t|z_{t-1}, a_{t-1})\right)\right], \quad (13)$$

where we omit subscripts $\theta$ and $\phi$, and a more detailed derivation can be found in Appendix B. Obviously, Equation (13) is differentiable with respect to the parameters of the model $\theta$ and $\phi$. Using the reparametrization trick (Kingma & Welling, 2013), we can directly apply backpropagation to update the parameters.

### 3.4.2 INFERENCE MODEL

From the factorization form in Equation (12), we can see that there are two types of inference models: $q(u|\vec{x}, \vec{a}, \vec{r})$ and $q(\vec{z}|\vec{x}, \vec{a}, \vec{r})$. Similar to the generative model in Section 3.2, we also parametrize both of them as Gaussian:

$$q(u|\vec{x}, \vec{a}, \vec{r}) = \mathcal{N}\left(u|\hat{\mu}_t^u, \hat{\sigma_t^u}^2\right); \quad \hat{\mu}_t^u = f_9(\vec{x}, \vec{a}, \vec{r}) \quad \hat{\sigma_t^u}^2 = f_{10}(\vec{x}, \vec{a}, \vec{r}); \quad (14)$$

$$q(\vec{z}|\vec{x}, \vec{a}, \vec{r}) = \mathcal{N}\left(\vec{z}|\hat{\mu}_t^{\vec{z}}, \hat{\sigma_t^{\vec{z}}}^2\right); \quad \hat{\mu}_t^{\vec{z}} = f_{11}(\vec{x}, \vec{a}, \vec{r}) \quad \hat{\sigma_t^{\vec{z}}}^2 = f_{12}(\vec{x}, \vec{a}, \vec{r}). \quad (15)$$

In fact, as shown in Equation (12), $q(\vec{z}|\vec{x}, \vec{a}, \vec{r})$ can be further factorized as the product of $q(z_t|z_{t-1}, \vec{x}, \vec{a}, \vec{r})$ for $t = 1, \ldots, T$. Taking a closer look at this term, based on the Markov property of our model, we have $z_t \perp\!\!\!\perp x_1, \ldots, x_{t-1}, a_1, \ldots, a_{t-2}, r_2, \ldots, r_t|z_{t-1}$, and then the term can be simplified as follows,

$$q(z_t|z_{t-1}, \vec{x}, \vec{a}, \vec{r}) = q(z_t|\underbrace{z_{t-1}, a_{t-1}}_{\text{past}}, \underbrace{x_t, a_t, r_{t+1}}_{\text{current}}, \underbrace{x_{t+1}, a_{t+1}, r_{t+2}, \ldots, x_T, a_T, r_{T+1}}_{\text{future}}). \quad (16)$$

Equation (16) tells us that $z_t$ depends on $z_{t-1}$ and all the current and future observed data $(\vec{x}, \vec{a}, \vec{r})$. Meanwhile, the conditional independence above means that $z_{t-1}$ contains all the historical data. Therefore, it is natural to calculate $z_t$ based on the whole sequence of data, which is exactly what recurrent neural networks (RNNs) do. Inspired by (Krishnan et al., 2015; 2017), we similarly choose a bi-directional LSTM (Zaremba & Sutskever, 2014) to parameterize $f_{11}$ and $f_{12}$ in Equation (15). Considering Equation (14) has the same structure as Equation (15), $f_9$ and $f_{10}$ are parameterized by a bi-directional LSTM as well. More details about the architecture can be found in Appendix D.

Note that for the task of sample predictions, that is, at any time step $t$, given a new $x_t$, we require to know $a_t$ and $r_{t+1}$ before inferring the distribution over $z_t$. Hence, we need to introduce two auxiliary distributions, denoted by red and blue dashed lines in Figure 1(b), to help conduct counterfactual reasoning (i.e., sample prediction on unseen $x_t$). To be more precise, we have

$$q(a_t|x_t) = \mathcal{N}\left(\mu = \hat{\mu}_t^{\mathfrak{a}}, \sigma^2 = \hat{\sigma_t^{\mathfrak{a}}}^2\right) \quad \hat{\mu}_t^{\mathfrak{a}} = f_{13}(x_t) \quad \hat{\sigma_t^{\mathfrak{a}}}^2 = f_{14}(x_t); \quad (17)$$

$$q(r_{t+1}|x_t, a_t) = \mathcal{N}\left(\mu = \hat{\mu}_t^{\mathfrak{r}}, \sigma^2 = \hat{\sigma_t^{\mathfrak{r}}}^2\right) \quad \hat{\mu}_t^{\mathfrak{r}} = f_{15}(x_t, a_t) \quad \hat{\sigma_t^{\mathfrak{r}}}^2 = f_{16}(x_t, a_t), \quad (18)$$

where $f_{13}$, $f_{14}$, $f_{15}$, and $f_{16}$ are also parameterized by neural networks. To estimate the parameters, we will add these two extra terms in the variational lower bound (Equation (13)):

$$\mathcal{L}_{\text{DRL}} = \mathcal{L} + \sum_{t=1}^{T} \left(\log q(a_t|x_t) + \log q(r_{t+1}|x_t, a_t)\right). \quad (19)$$

### 3.5 DECONFOUNDING RL ALGORITHMS

Now we have all the building blocks for DRL algorithms. Once our model is learned from the observational data, it can be directly used as a dynamic environment like those in OpenAI Gym (Brockman et al., 2016). We can exploit the learned model to generate rollouts for policy optimization. In practice, Equation (8) is approximated using the Monte Carlo method as follows:

$$p(r_t|z_t = \mathfrak{z}, \text{do}(a_t = \mathfrak{a})) = \frac{1}{N} \sum_{i=1}^{N} p(r_t|z_t = \mathfrak{z}, a_t = \mathfrak{a}, u_i) \quad u_i \sim p(u), \quad (20)$$

where $N$ is the number of samples from the prior $p(u)$. In the presence of observational data, a better estimate could be given on the basis of the samples drawn from the approximate posterior $q(u|\vec{x}, \vec{a}, \vec{r})$ which we compute through the inference network presented in Section 3.4.2.

On the basis of our deconfounding reward function, it is straightforward to extend traditional RL algorithms to their corresponding deconfounding version. In this paper, we select one representative of them: the Actor-Critic method (Sutton et al., 1998), but it is straightforward to be applied to other algorithms.

**Deconfounding Actor-Critic Methods**    The actor-critic method is a policy-based method directly parameterizing the policy $\pi(a|z; \theta)$, which aims to reduce the variance of the estimate of the policy gradient by subtracting a learned function of the state $b(z)$, known as a baseline, from the return. The learned value function $V(z; \phi)$ is commonly used as the baseline. Taking into consideration that the return is the estimate of $Q(z, a; \phi_Q)$ and $b(z)$ is the estimate of $V(z; \phi_V)$, the gradient of the actor-critic loss function at step time $t$ is given by

$$\nabla J(\theta) = \mathbb{E}_\pi \left[ (Q(z_t, a_t; \phi_Q) - V(z_t; \phi_V)) \nabla_\theta \ln \pi(a_t|z_t; \theta) \right], \tag{21}$$

where $Q(z_t, a_t; \phi_Q) - V(z_t; \phi_V)$ used to be seen as an estimate of the advantage of action $a_t$ in state $z_t$. In practice, $Q(z_t, a_t; \phi_Q)$ is usually replaced with one-step return, that is, $r_{t+1} + V(z_{t+1}; \phi_V)$. Importantly, in deconfounding actor-critic methods, we use $r_{t+1} \sim p(r_{t+1}|z_t, \text{do}(a_t))$ (Equation (20)), as opposed to $r_{t+1} \sim p(r_{t+1}|z_t, a_t)$ in vanilla actor-critic methods. The details about training can be found in Appendix E.

## 4    EXPERIMENTAL RESULTS

It is widely acknowledged that evaluating approaches dealing with confounding is always challenging due to lack of groundtruth and benchmark datasets. Besides, little work has been done before in DRL, which renders evaluating such algorithms in this respect much harder. Therefore, we first develop benchmark datasets for evaluation of our algorithms, primarily by revising the MNIST dataset (LeCun et al., 1998) and two environments in OpenAI Gym (Brockman et al., 2016): CartPole and Pendulum. We then evaluate our model as well as compare two proposed deconfounding algorithms to their corresponding vanilla versions on the benchmark datasets we developed.

### 4.1    IMPLEMENTATION DETAILS

We used Tensorflow (Abadi et al., 2016) for the implementation of our model and DRL algorithms. Optimization was done with Adam (Kingma & Ba, 2014). Unless stated otherwise, the setting of all the hyperparameters and architectures of the neural networks we adopted in this paper can be found in Appendix J.

To verify how good the learned model is, we performed two types of tasks: reconstruction and counterfactual reasoning. The reconstructions were performed by feeding the input sequence into the learned inference network, and then sampling from the resulting posterior distribution according to Equation (15), and finally feeding those samples into the generative network described in Equation (3). The counterfactual reasoning, that is, predicting $x_{t+1}$ given $x_t$ we have not seen in the training set and a model trained in the training set, were executed through four steps: 1) Given an $x_t$ unseen in the training set, we estimate $a_t$ and $r_{t+1}$ based on Equation (17) and (18); 2) Once we have $x_t$, $a_t$, and $r_{t+1}$, it is easy to estimate $z_t$ from Equation (15); 3) Using the estimated $z_t$ and $a_t$, we can directly compute $z_{t+1}$ from Equation (6); 4) The final step is to reconstruct $x_{t+1}$ from $z_{t+1}$ and $u$ according to Equation (3). Repeating the four steps, we can counterfactually reason out a sequence of data.

To evaluate the confounder $u$, there are also two scenarios. The easy one is that, given a sequence of observational data $(\vec{x}, \vec{a}, \vec{r})$, it is obvious to estimate $u$ from Equation (14). The more challenging one is to calculate $u$ given only $x_t$ at any time step. Following the same steps used in the task of counterfactual reasoning, we first compute $a_t$ and $r_{t+1}$ based on Equation (17) and (18), and then estimate $u$ through Equation (14).

### 4.2    CONFOUNDING DATASETS

Due to the space limit, we only introduce the *Confounding Pendulum* dataset in this section. *Confounding CartPole* and *Confounding MNIST* are devised in the same way and can be found in Appendix H.1 and Appendix H.2, respectively.

**Confounding Pendulum**    Motivated by (Krishnan et al., 2015) where the authors synthesized a dataset mimicking the healthcare data under harsh conditions (e.g., noisy laboratory measurements, surgeries and drugs affected by patient age and gender, etc.), as well as considering our focus on RL tasks, we revised the original Pendulum in OpenAI Gym (Brockman et al., 2016) to develop a confounding Pendulum dataset in which a binary confounder is introduced. More specifically, we select 100 different screen images of Pendulum to create a synthetic dataset where actions are joint effort with the value of between $-2$ and $2$ [4]. First, $20\%$ bit-flip noise are added to each image , and then, based on a random policy but confounded by a binary factor (will be described soon), actions are performed on each noisy image for five time steps, which produces a large number of 5-step sequences of noisy images. To each generated sequence, exactly one sequence of three consecutive squares ($2 \times 2$ in pixel) is superimposed with the top-left corner of the images in a random starting location. The squares within the sequences are intended to be analogous to seasonal flu or other ailments that a patient could exhibit that are independent of the actions and which last several time steps (Krishnan et al., 2015). We aim to show that our model can learn long-range patterns, which plays an important role in medical applications. We treat such generated sequences of images as the observations $\vec{x}$. The training/validation/test set respectively comprises $140000/28000/28000$ sequences of length five.

Now we are arriving at the key stage: how to define a reasonable relationship between the confounder, action, and reward. For simplicity of notation, we denote the confounder by $u$, action by $a$, and reward by $r$. $u$ is a binary variable mimicking the socio-economic status (i.e., the rich and the poor) . The range of actions $a \in [-2, 2]$ is grouped into two categories $T_1$ (i.e., $1 \leq |T_1| \leq 2$) and $T_2$ (i.e., $0 \leq |T_2| \leq 1$) representing different treatments (i.e., $T_1$ is more valid and $T_2$). How to choose the treatment depends on $u$. The reward is defined as follows,

$$r = r_{\mathrm{o}} + r_{\mathrm{c}}, \tag{22}$$

where $r_o$ is the original reward in Pendulum, which is a function of $a$ [5], and $r_c$ is the extra reward caused by both $u$ and $a$, where $r_c$ comes from a two Gaussian mixture

$$r_c \sim \sum\nolimits_{r_c = \{R_1, R_2\}} \pi_{r_c} \mathcal{N}(\mu_{r_c}, \sigma^2), \quad \sum\nolimits_{r_c = \{R_1, R_2\}} \pi_{r_c} = 1, \pi_{r_c = R_1}, \pi_{r_c = R_2} \geq 0, \tag{23}$$

with $\sigma$ and $\mu_{r_c}$ fixed and $\pi_{r_c}$ determined by both $u$ and $a$ (see more in Appendix H.3). Obviously, in the definition above, $r$ is a function of $a$ and $u$. It is worth noting that $u$ has an influence on $x_t$ through $a_{t-1}$ and $z_t$, meaning that $x_t$ contains some piece of information about $u$ and therefore can be viewed as its proxy variable. In our case, for the sake of generality and practice, we assume the influence between the confounder, action and reward is stochastic. Take the case of kidney stones for example, even though treatment $a$ is more valid than treatment $b$, there are still some patients choosing treatment $b$ in each category, but with different probabilities. All the details can be found in Appendix H.3, where a straightforward analogy is provided as well.

### 4.3    PERFORMANCE ANALYSIS OF THE DECONFOUNDING MODEL

In this section, to demonstrate the validity of our deconfounding model, denoted by $M_{\mathrm{decon}}$, we compare with the original model (i.e., the model similar to that shown in Figure 1(b) but without the confounder $u$), denoted by $M_{\mathrm{orin}}$. We train $M_{\mathrm{decon}}$ by optimizing Equation (19) but train $M_{\mathrm{orin}}$ by a little different loss function excluding the confounder $u$ whose full derivation can be found in Appendix C. Both models are separately trained in a batch manner on the training set (i.e., $140K$ sequences of length five of images) of the confounding dataset. Afterwards, following the steps depicted in Section 4.1, we use each trained model to perform the reconstruction task on the training set, and both reconstruction and counterfactual reasoning tasks on the testing set (i.e., $28K$ sequences of length five of images).

Figure 2 presents a comparison of $M_{\mathrm{decon}}$ and $M_{\mathrm{orin}}$, in terms of reconstruction and counterfactual reasoning on the confounding Pendulum dataset. The second row is based on $M_{\mathrm{decon}}$ (Figure 1(b)), whilst the top row comes from $M_{\mathrm{orin}}$. It is evident that the results generated by the deconfounding model is superior to those produced by the model not taking into account the confounder. To be more specific, as shown in the zoom of samples on the bottom row, $M_{\mathrm{orin}}$ generates more blurry images than $M_{\mathrm{decon}}$, because, without modelling the confounder $u$, $M_{\mathrm{orin}}$ is forced to average over its

---

[4]More details can be found at `https://github.com/openai/gym/wiki/Pendulum-v0`.

[5]In fact, here $r_o$ is a function of both $a$ and a state, and we mention only $a$ to emphasize that the confounder affects the action.

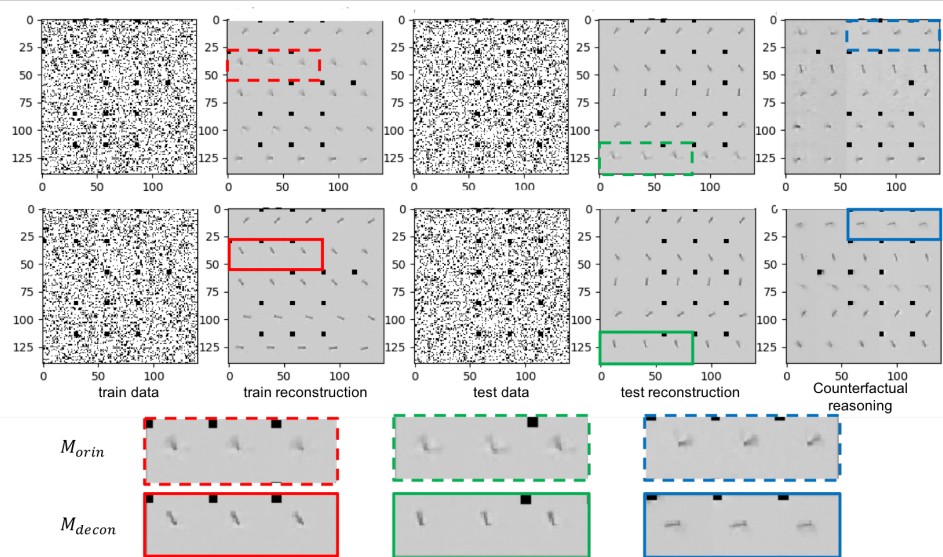

Figure 2: Reconstruction and counterfactual reasoning on the confounding Pendulum dataset. Top row: results from the model without the confounder $u$ ($M_{\text{orin}}$); Second row: results from the model with the confounder $u$ ($M_{\text{decon}}$). The last two rows are the zoom of samples selected in the same positions from the top row (dashed boxes for $M_{\text{orin}}$) and the second row (solid boxes for $M_{\text{decon}}$), respectively.

multiple latent states resulting in more blurry samples. Likewise, we can attain the same conclusion from the samples respectively produced by $M_{\text{orin}}$ and $M_{\text{decon}}$ on the confounding MNIST dataset, as shown in Figure 7 of Appendix I.

In addition, looking closely at the squares on the generated digit samples (colored in yellow box), it is obvious to observe that $M_{\text{orin}}$ will generate non-consecutive squares in the task of counterfactual reasoning, which does not really make sense because only consecutive patterns appear in the training set. In contrast, this case does not take place on the samples from $M_{\text{decon}}$, showing that our deconfounding model is able to cope with long-range patterns.

It is worth noting that, as shown in Figure 8 of Appendix I, by visualizing the 2-dimensional confounder $u$, we can discern that although the prior distribution of the confounder is assumed to be a unit Gaussian distribution, the model still can learn two obvious clusters from the data because it is originally a binary variable. It demonstrates that our model has an advantage in learning confounders even if the assumed prior over them were not that accurate.

## 4.4 COMPARISON OF RL ALGORITHMS

In this section, we will evaluate the proposed deconfounding actor-critic (AC) method by comparing with its vanilla version on the confounding Pendulum dataset. In the vanilla AC method, given a learned $M_{\text{orin}}$, we optimize the policy by calculating the gradient presented in Equation (21) on the basis of the trajectories/rollouts generated through $M_{\text{orin}}$. Equation (21) involves two functions: $V(z_t; \phi_V)$ and $\pi(a_t|z_t; \theta)$, whose parameters can be found in Appendix J. It is worth noting that, in this vanilla case, each reward $r_{t+1}$ is produced from the conditional distribution $p(r_{t+1}|z_t, a_t)$. In contrast, the proposed deconfounding AC method is built on $M_{\text{decon}}$. Although the same gradient method (Equation (21)) is utilized to optimize the policy, we base the deconfounding AC approach on the different trajectories/rollouts generated by $M_{\text{decon}}$ in which each reward $r_{t+1}$ relies on the interventional distribution $p(r_{t+1}|z_t, \text{do}(a_t))$ computed using Equation (20).

In the training phase, for both vanilla AC and deconfounding AC, we run a respective experiment over 1500 episodes with 200 time steps each. In order to reduce non-stationarity and to decorrelate updates, the generated data is stored in an experience replay memory and then randomly sampled in a batch manner (Mnih et al., 2013; Riedmiller, 2005; Schulman et al., 2015; Van Hasselt et al., 2016). In each episode, we summarize all the rewards and further average the sums over a window of 100 episodes to obtain a more smooth curve. As shown in Figure 3(a), obviously our deconfounding AC algorithm performs significantly better than the vanilla AC algorithm in the confounded environment.

In the testing phase, we first randomly select 100 samples from the testing set, each starting a new episode, and then use the learned policies to perform reasoning over 200 time steps as we did during the training time. From the resulting 100 episodes, we plot the total reward for each, shown in Figure 3(b), and compute the percentage of the optimal action $T_1$ in each episode, presented in Figure 3(c). It is worth noting that Figure 3(c) tells us that in each episode our deconfounding AC almost chooses the optimal action at each time step, whilst the vanilla AC makes a wrong decision for more than half time.

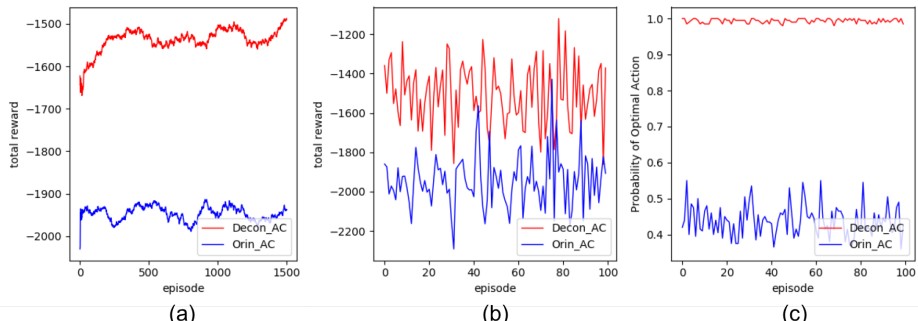

Figure 3: Comparison of vanilla (Orin_AC) and deconfounding (Decon_AC) Actor-Critic methods on the confounding Pendulum dataset. (a) total reward over 1500 episodes in the training phase; (b) total reward over 100 episodes in the testing phase; (c) Probability of optimal action over 100 episodes, corresponding to (b).

## 5 RELATED WORK

Krishnan et al. (2015; 2017) used deep neural networks to model nonlinear state space models and leveraged a structured variational approximation parameterized by recurrent neural networks to mimic the posterior distribution. Levine (2018) reformulated RL and control problems to probabilistic inference, which allows us to bear a large pool of approximate inference methods, and flexibly extend the model. Raghu et al. (2017a;b) exploited continuous state-space models and deep RL to deduce treatment policies for septic patients from observational data. Gottesman et al. (2018) discussed some issues of evaluating RL algorithms in observational health setting. However, all the work mentioned above did not take into account confounders in their models.

Louizos et al. (2017) attempted to learn individual-level causal effects from observational data using variational auto-encoder to estimate the unknown confounder given a causal graph in then non-temporal setting. Paxton et al. (2013) developed predictive models based on electronic medical records without using causal inference. Saria et al. (2010) proposed a nonparametric Bayesian method to analyze clinical temporal data. Soleimani et al. (2017) represented the treatment response curves using linear time-invariant dynamical systems which provides a flexible approach to modeling response over time. Although the latter two work modeled the sequential data, they both do not exploit RL or causal inference.

Bareinboim et al. (2015) considered the problem of bandits with unobserved confounders, which is one quite simple RL setting without state transitions. Sen et al. (2016) and Ramoly et al. (2017) further studied contextual bandits with latent confounders. Forney et al. (2017) circumvented some problems caused by unobserved confounders in Multi-Armed Bandit by counterfactual-based decision-making. Zhang & Bareinboim (2017) leveraged causal inference to tackle the problem of transferring knowledge across bandit agents. However, all these methods are based on the bandit problem, a simplified version of RL, instead of the full RL problem.

In fact, as far as we are concerned, this is the first attempt to build a bridge between confounding and the full RL problem, and this is also one of few research papers aiming at understanding the connections between causal inference and full RL.

## 6 CONCLUSION AND FUTURE WORK

To address the confounding issue in RL, we introduced a general formulation, namely deconfounding reinforcement learning. On the basis of the proposed formulation, we presented deconfounding variants of actor-critic methods and showed their superior performance on confounding datasets that we created by revising OpenAI Gym and MNIST. In the future, we will collaborate with hospitals and apply our approach to real-world medical datasets. We also hope that our work will stimulate further investigation of connections between causal inference and RL.

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

# Appendices

## A    IDENTIFICATION OF CAUSAL EFFECT

As pointed out in (Louizos et al., 2017; Edwards et al., 2015; Kuroki & Pearl, 2014; Miao et al., 2018; Pearl, 2012; Wooldridge, 2009), it is impossible to recover the joint distribution from the observational data if the hidden confounder has no connection to the observed variables. Fortunately, there still exist a large number of possible circumstances, and we refer readers to (Louizos et al., 2017; Kuroki & Pearl, 2014; Miao et al., 2018; Pearl, 2012; Allman et al., 2009) for more details. Here, we only focus on one possible case presented in Figure 1 of (Louizos et al., 2017) (see Appendix F.1), because the result can be directly used to identify the causal effect in our model. More precisely, in our model, at each time step, the 4-tuple $(z_t, x_t, a_t, r_{t+1})$ is exactly the same as that case in Figure 1 of (Louizos et al., 2017), and therefore $p(z_t, x_t, a_t, r_{t+1})$ can be approximately recovered only from the observations $(x_t, a_t, r_{t+1})$. Likewise, the joint distribution over the other 4-tuple $(u, x_t, a_t, r_{t+1})$ can be recovered from $(x_t, a_t, r_{t+1})$ in the same manner (see Appendix G). Applying this rule repeatedly to the sequential data, we are finally able to approximately recover $p(u, \vec{z}, \vec{x}, \vec{a}, \vec{r})$ solely from the observations $(\vec{x}, \vec{a}, \vec{r})$.

We recover the joint distribution using Variational Auto-Encoders (VAEs) (Kingma & Welling, 2013), which is a powerful tool to recover a very large class of latent-variable models as a minimizer of an optimization problem (Tran et al., 2015; Louizos et al., 2017). Generally speaking, however, VAEs currently have little theory available to guarantee that the true model can be identified through learning (Louizos et al., 2017). The main reason behind is that, in such latent variable models, confounders and causal parameters, both of which are unknown and might influence each other, will inevitably result in the case that different causal parameters could generate the same observable data (D'Amour, June 9, 2018). Fortunately, identification is possible when the latent confounder and their relationship to the outcome are themselves identified. For instance, in the presence of proxy variables of the latent confounder, as described in the previous paragraph, both our model and the one in (Louizos et al., 2017) are identified under some conditions outlined in (Louizos et al., 2017; Kuroki & Pearl, 2014; Miao et al., 2018; Pearl, 2012; Allman et al., 2009). In addition, even if only considering our model, because of the existence of many proxy variables of the confounders (e.g., each of $z_t$ and $u$ has at least $T$ proxy variables $\{x_t\}_{t=1,...,T}$, where $z_t$ can be viewed as hidden states in HMMs (Allman et al., 2009)), identification can be also justified if the latent confounders $z_t$ and $u$ are categorical (Allman et al., 2009). In practice, however, we are not able to know the exact nature of the confounders, e.g., what kinds of distributions they follow, if categorical how many categories they have, etc. Hence, we take advantage of VAEs, which allows us to make substantially weaker assumptions about the data generating process and the structure of the hidden confounders (Louizos et al., 2017). Nevertheless, we empirically show that our approach is more beneficial to learning a better policy in the existence of confounders.

Note that, in our model we have to differentiate the two types of confounders: the time-independent confounder $u$ and time-dependent confounders $\{z_t\}_{t=1,...,T}$, each playing a respective role in the model. The former, as a global confounder, will affect the whole course of treatment, and therefore should be adjusted for. In the example of kidney stones, the existence of the confounder (i.e., the size of stones) will lead to a wrong treatment if not adjusting for it. In contrast, the time-varying confounders $\{z_t\}$ act as states in RL, which, in principle, should not be adjusted for, because the goal in RL is to learn a good policy in which any action is indeed supposed to be conditional on a specific state. On the other hand, in terms of rewards, what an agent expects at each time step is exactly the immediate reward when taking a specific action at a specific state, without the need of adjusting for states. It is worth noting that the case with time-varying confounders $\{z_t\}$ can be thought of to meet a *pseudo or weak causal sufficiency* assumption under which the causal effects of actions on rewards will not be influenced by states at each time step (Zhang et al., 2017; 2015). This key difference motivates us to only adjust for the time-independent confounder $u$ in this paper. In addition, as shown in Figure 1(b), in our case where a policy depending only on the confounder is applied to generating the data (Section 4.2), the arrow from $z_t$ to $a_t$ is not necessary so that $z_t$ can be viewed as not a confounder of $a_t$ and $r_{t+1}$. This also provides another reason why we do not need to adjust for $z_t$.

# B  Variational Lower Bound for $M_{\text{DECON}}$

$$\log p_\theta(\vec{x}, \vec{a}, \vec{r})$$

$$= \log \int_u \int_{\vec{z}} p_\theta(\vec{x}, \vec{a}, \vec{r}, \vec{z}, u) \mathrm{d}\vec{z}\mathrm{d}u$$

$$\geq \int_u \int_{\vec{z}} q_\phi(\vec{z}, u|\vec{x}, \vec{a}, \vec{r}) \log \frac{p_\theta(\vec{x}, \vec{a}, \vec{r}, \vec{z}, u)}{q_\phi(\vec{z}, u|\vec{x}, \vec{a}, \vec{r})} \mathrm{d}\vec{z}\mathrm{d}u$$

$$= \int_u \int_{\vec{z}} q_\psi(u|\vec{x}, \vec{a}, \vec{r}) q_\phi(\vec{z}|\vec{x}, \vec{a}, \vec{r}) \log \frac{p_\theta(\vec{x}, \vec{a}, \vec{r}, \vec{z}, u)}{q_\psi(u|\vec{x}, \vec{a}, \vec{r}) q_\phi(\vec{z}|\vec{x}, \vec{a}, \vec{r})} \mathrm{d}\vec{z}\mathrm{d}u \qquad \text{(factorization assumption)}$$

$$= \int_u \int_{\vec{z}} q_\psi(u|\vec{x}, \vec{a}, \vec{r}) q_\phi(\vec{z}|\vec{x}, \vec{a}, \vec{r})$$
$$\log \frac{p(u)p(z_1) \left[\prod_{t=1}^T p(x_t|z_t, u)p(a_t|z_t, u)p(r_{t+1}|z_t, a_t, u)\right] \left[\prod_{t=2}^T p(z_t|z_{t-1}, a_{t-1})\right]}{q_\psi(u|\vec{x}, \vec{a}, \vec{r}) q_\phi(\vec{z}|\vec{x}, \vec{a}, \vec{r})} \mathrm{d}\vec{z}\mathrm{d}u$$

$$= \int_u \int_{\vec{z}} q(u|\vec{x}, \vec{a}, \vec{r}) q(z_1|\vec{x}, \vec{a}, \vec{r}) \cdots q(z_T|z_{T-1}, \vec{x}, \vec{a}, \vec{r})$$
$$\log \frac{p(u)p(z_1) \prod_{t=1}^T p(x_t|z_t, u)p(a_t|z_t, u)p(r_{t+1}|z_t, a_t, u) \prod_{t=2}^T p(z_t|z_{t-1}, a_{t-1})}{q(u|\vec{x}, \vec{a}, \vec{r}) q(z_1|\vec{x}, \vec{a}, \vec{r}) \cdots q(z_T|z_{T-1}, \vec{x}, \vec{a}, \vec{r})} \mathrm{d}\vec{z}\mathrm{d}u$$

$$= \sum_{t=1}^T \int_u \int_{z_1} \cdots \int_{z_T} q(u|\vec{x}, \vec{a}, \vec{r}) q(z_1|\vec{x}, \vec{a}, \vec{r}) \cdots q(z_T|z_{T-1}, \vec{x}, \vec{a}, \vec{r})$$
$$\log \left( p(x_t|z_t, u)p(a_t|z_t, u)p(r_{t+1}|z_t, a_t, u) \right) \mathrm{d}\vec{z}\mathrm{d}u$$

$$+ \int_u \int_{z_1} \cdots \int_{z_T} q(u|\vec{x}, \vec{a}, \vec{r}) q(z_1|\vec{x}, \vec{a}, \vec{r}) \cdots q(z_T|z_{T-1}, \vec{x}, \vec{a}, \vec{r}) \log \frac{p(u)}{q(u|\vec{x}, \vec{a}, \vec{r})} \mathrm{d}\vec{z}\mathrm{d}u$$

$$+ \int_u \int_{z_1} \cdots \int_{z_T} q(u|\vec{x}, \vec{a}, \vec{r}) q(z_1|\vec{x}, \vec{a}, \vec{r}) \cdots q(z_T|z_{T-1}, \vec{x}, \vec{a}, \vec{r}) \log \frac{p(z_1)}{q(z_1|\vec{x}, \vec{a}, \vec{r})} \mathrm{d}\vec{z}\mathrm{d}u$$

$$+ \sum_{t=2}^T \int_u \int_{z_1} \cdots \int_{z_T} q(u|\vec{x}, \vec{a}, \vec{r}) q(z_1|\vec{x}, \vec{a}, \vec{r}) \cdots q(z_T|z_{T-1}, \vec{x}, \vec{a}, \vec{r}) \log \frac{p(z_t|z_{t-1}, a_{t-1})}{q(z_t|z_{t-1}, \vec{x}, \vec{a}, \vec{r})} \mathrm{d}\vec{z}\mathrm{d}u$$

$$= \sum_{t=1}^T \int_u \int_{z_t} q(u|\vec{x}, \vec{a}, \vec{r}) q(z_t|z_{t-1}, \vec{x}, \vec{a}, \vec{r}) \log \left( p(x_t|z_t, u)p(a_t|z_t, u)p(r_{t+1}|z_t, a_t, u) \right) \mathrm{d}z_t\mathrm{d}u$$

$$+ \int_u q(u|\vec{x}, \vec{a}, \vec{r}) \log \frac{p(u)}{q(u|\vec{x}, \vec{a}, \vec{r})} \mathrm{d}u$$

$$+ \int_{z_1} q(z_1|\vec{x}, \vec{a}, \vec{r}) \log \frac{p(z_1)}{q(z_1|\vec{x}, \vec{a}, \vec{r})} \mathrm{d}z_1$$

$$+ \sum_{t=2}^T \int_{z_{t-1}} \int_{z_t} q(z_t|z_{t-1}, \vec{x}, \vec{a}, \vec{r}) \log \frac{p(z_t|z_{t-1}, a_{t-1})}{q(z_t|z_{t-1}, \vec{x}, \vec{a}, \vec{r})} \mathrm{d}z_t\mathrm{d}z_{t-1}$$

$$= \sum_{t=1}^T \mathop{\mathbb{E}}_{\substack{z_t \sim q(z_t|z_{t-1}, \vec{x}, \vec{a}, \vec{r}) \\ u \sim q(u|\vec{x}, \vec{a}, \vec{r})}} \left[ \log p(x_t|z_t, u) + \log p(a_t|z_t, u) + \log p(r_{t+1}|z_t, a_t, u) \right]$$

$$- \text{KL}\left( q(u|\vec{x}, \vec{a}, \vec{r}) || p(u) \right)$$
$$- \text{KL}\left( q(z_1|\vec{x}, \vec{a}, \vec{r}) || p(z_1) \right)$$
$$- \sum_{t=2}^T \mathop{\mathbb{E}}_{z_{t-1} \sim q(z_{t-1}|z_{t-2}, \vec{x}, \vec{a}, \vec{r})} \left[ \text{KL}\left( q(z_t|z_{t-1}, \vec{x}, \vec{a}, \vec{r}) || p(z_t|z_{t-1}, a_{t-1}) \right) \right].$$

## C    VARIATIONAL LOWER BOUND FOR $M_{\text{ORIN}}$

$$\log p_\theta(\vec{x}, \vec{a}, \vec{r})$$

$$= \log \int_{\vec{z}} p_\theta(\vec{x}, \vec{a}, \vec{r}, \vec{z}) \mathrm{d}\vec{z}$$

$$\geq \int_{\vec{z}} q_\phi(\vec{z}|\vec{x}, \vec{a}, \vec{r}) \log \frac{p_\theta(\vec{x}, \vec{a}, \vec{r}, \vec{z})}{q_\phi(\vec{z}|\vec{x}, \vec{a}, \vec{r})} \mathrm{d}\vec{z}$$

$$= \int_{\vec{z}} q_\phi(\vec{z}|\vec{x}, \vec{a}, \vec{r}) \log \frac{p(z_1) \left[ \prod_{t=1}^{T} p(x_t|z_t)p(a_t|z_t)p(r_{t+1}|z_t, a_t) \right] \left[ \prod_{t=2}^{T} p(z_t|z_{t-1}, a_{t-1}) \right]}{q_\phi(\vec{z}|\vec{x}, \vec{a}, \vec{r})} \mathrm{d}\vec{z}$$

$$= \int_{\vec{z}} q(z_1|\vec{x}, \vec{a}, \vec{r}) \cdots q(z_T|z_{T-1}, \vec{x}, \vec{a}, \vec{r})$$

$$\log \frac{p(z_1) \prod_{t=1}^{T} p(x_t|z_t)p(a_t|z_t)p(r_{t+1}|z_t, a_t) \prod_{t=2}^{T} p(z_t|z_{t-1}, a_{t-1})}{q(z_1|\vec{x}, \vec{a}, \vec{r}) \cdots q(z_T|z_{T-1}, \vec{x}, \vec{a}, \vec{r})} \mathrm{d}\vec{z}$$

$$= \sum_{t=1}^{T} \int_{z_1} \cdots \int_{z_T} q(z_1|\vec{x}, \vec{a}, \vec{r}) \cdots q(z_T|z_{T-1}, \vec{x}, \vec{a}, \vec{r}) \log \left( p(x_t|z_t)p(a_t|z_t)p(r_{t+1}|z_t, a_t) \right) \mathrm{d}\vec{z}$$

$$+ \int_{z_1} \cdots \int_{z_T} q(z_1|\vec{x}, \vec{a}, \vec{r}) \cdots q(z_T|z_{T-1}, \vec{x}, \vec{a}, \vec{r}) \log \frac{p(z_1)}{q(z_1|\vec{x}, \vec{a}, \vec{r})} \mathrm{d}\vec{z}$$

$$+ \sum_{t=2}^{T} \int_{z_1} \cdots \int_{z_T} q(z_1|\vec{x}, \vec{a}, \vec{r}) \cdots q(z_T|z_{T-1}, \vec{x}, \vec{a}, \vec{r}) \log \frac{p(z_t|z_{t-1}, a_{t-1})}{q(z_t|z_{t-1}, \vec{x}, \vec{a}, \vec{r})} \mathrm{d}\vec{z}$$

$$= \sum_{t=1}^{T} \int_{z_t} q(z_t|z_{t-1}, \vec{x}, \vec{a}, \vec{r}) \log \left( p(x_t|z_t)p(a_t|z_t)p(r_{t+1}|z_t, a_t) \right) \mathrm{d}z_t$$

$$+ \int_{z_1} q(z_1|\vec{x}, \vec{a}, \vec{r}) \log \frac{p(z_1)}{q(z_1|\vec{x}, \vec{a}, \vec{r})} \mathrm{d}z_1$$

$$+ \sum_{t=2}^{T} \int_{z_{t-1}} \int_{z_t} q(z_t|z_{t-1}, \vec{x}, \vec{a}, \vec{r}) \log \frac{p(z_t|z_{t-1}, a_{t-1})}{q(z_t|z_{t-1}, \vec{x}, \vec{a}, \vec{r})} \mathrm{d}z_t \mathrm{d}z_{t-1}$$

$$= \sum_{t=1}^{T} \mathbb{E}_{z_t \sim q(z_t|z_{t-1}, \vec{x}, \vec{a}, \vec{r})} \left[ \log p(x_t|z_t) + \log p(a_t|z_t) + \log p(r_{t+1}|z_t, a_t) \right]$$

$$- \mathrm{KL} \left( q(z_1|\vec{x}, \vec{a}, \vec{r}) || p(z_1) \right)$$

$$- \sum_{t=2}^{T} \mathbb{E}_{z_{t-1} \sim q(z_{t-1}|z_{t-2}, \vec{x}, \vec{a}, \vec{r})} \left[ \mathrm{KL} \left( q(z_t|z_{t-1}, \vec{x}, \vec{a}, \vec{r}) || p(z_t|z_{t-1}, a_{t-1}) \right) \right].$$

## D    BI-DIRECTIONAL LSTM

In our inference model, we use a similar architecture of bi-directional LSTM to that in (Krishnan et al., 2017). Apart from different inputs, the main difference is to introduce $a_{t-1}$ to computing the

combined hidden feature as shown in the following formula.

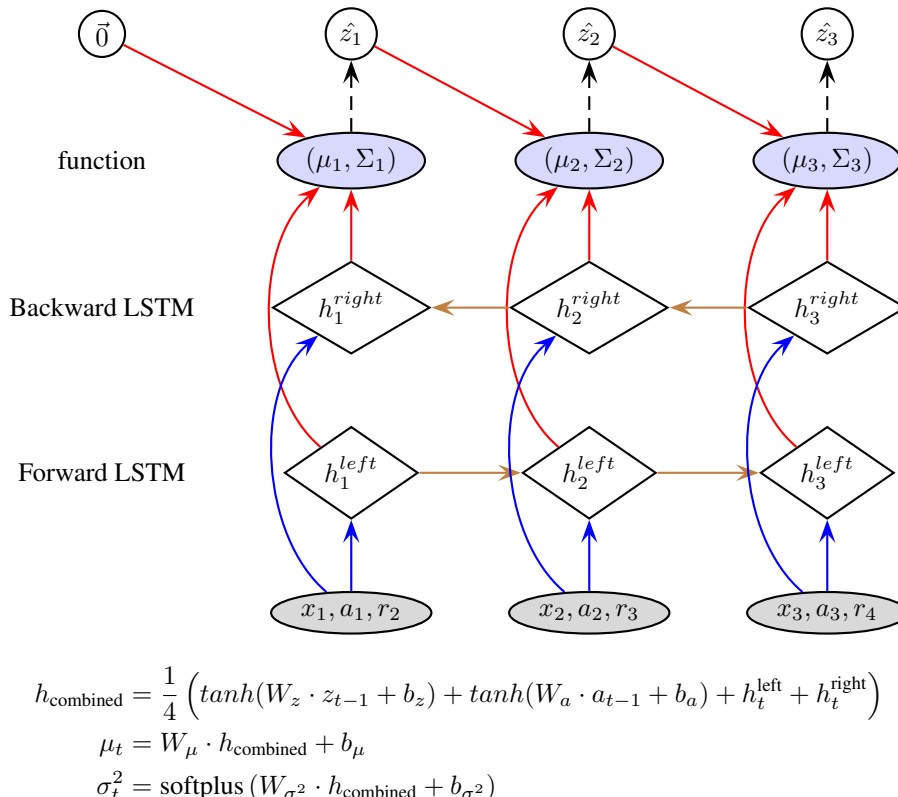

$$h_{\text{combined}} = \frac{1}{4} \left( tanh(W_z \cdot z_{t-1} + b_z) + tanh(W_a \cdot a_{t-1} + b_a) + h_t^{\text{left}} + h_t^{\text{right}} \right)$$
$$\mu_t = W_\mu \cdot h_{\text{combined}} + b_\mu$$
$$\sigma_t^2 = \text{softplus}\left( W_{\sigma^2} \cdot h_{\text{combined}} + b_{\sigma^2} \right)$$

## E  TRAINING DETAILS

As mentioned in the main body, DRL consists of two steps: learning a deconfounding model and optimizing a policy based on the learned deconfounding model. To be more specific, in step 1, given the observational data $(\vec{x}, \vec{a}, \vec{r})$, we learn the deconfounding model by optimizing the variational lower bound as presented in Equation (19). Once the deconfounding model is learned, we know the state transition function $p(z_t | z_{t-1}, a_{t-1})$ and can also calculate the deconfounding reward function $p(r_t | z_t, do(a_t))$ according to Equation (20). In step 2, we treat the learned deconfounding model as a RL environment like CartPole in OpenAI Gym, and directly exploit it to generate trajectories/rollouts through the state transition function and the deconfounding reward function. On the basis of the generated trajectories/rollouts, we can train the policy network using Equation (21).

## F  EXAMPLE OF KIDNEY STONES

### F.1  EXAMPLE OF A PROXY VARIABLE

For convenience, we directly move Figure 1 of (Louizos et al., 2017) here, as shown in Figure 4.

### F.2  INTUITION

Take for example the case of kidney stones, in the existence of the confounder (i.e., the size of stones), one prefers treatment $b$ when considering the overall probability of recovery, whilst one chooses with more probabilities treatment $a$ when investigating the recovery rate in each size. The correct answer is treatment $a$ in this example. The $do$-operator in Equation (2) of Section 2.3 provides a formal approach to seeking the correct solution to such confounded problems, which can not be addressed within the traditional RL framework, because the traditional framework only involves the conditional probability (e.g., the overall probability of recovery) instead of the interventional probability as $do$-operator does.

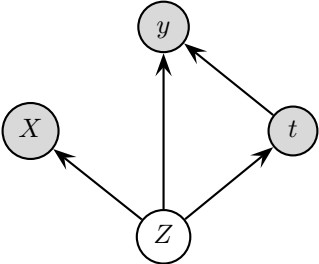

Figure 4: Example of a proxy variable. $t$ is a treatment, e.g., medication; $y$ is an outcome, e.g., mortality; $Z$ is an unobserved confounder, e.g., socio-economic status; and $X$ is noisy views on the hidden confounder $Z$, say income in the last year and place of residence.

## G  ANALYSIS OF THE DECONFOUNDING MODEL

At each time step $t$, from the deconfounding model presented in Figure 1(b), we can extract two 4-tuple components, centering at $z_t$ (Figure 5) and $u$ (Figure 6), respectively. It is apparent to observe that both Figure 5 and Figure 6 share the exact same structure with Figure 4.

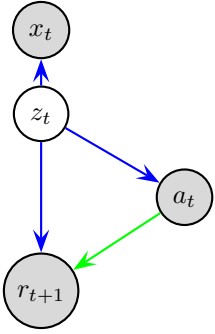

Figure 5: The component of 4-tuple $(z_t, x_t, a_t, r_{t+1})$.

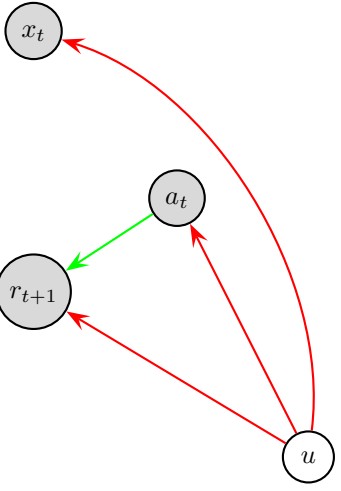

Figure 6: The component of 4-tuple $(u, x_t, a_t, r_{t+1})$.

## H  CONFOUNDING DATASETS

### H.1  CONFOUNDING CARTPOLE

The confounding CartPole dataset can be implemented in the exact same manner as Confounding Pendulum, except for the action which is originally binary and can be naturally divided into two categories.

### H.2  CONFOUNDING MNIST

We follow the same setting to develop a confounding MNIST dataset, but the definitions of action and the original reward term are different. Similar to the Healing MNIST dataset (Krishnan et al., 2015), rotations of digit images are encoded as the actions $\vec{a}$ $(-45 \leq a \leq 45)$ which, according to the binary confounder $u$, is divided into two categories $(22.5 \leq |a| \leq 45)$ and $(0 \leq |a| < 22.5)$. The original reward term $r_o$ is defined as the minus degree between the upright position and the position the digit rotates to. For example, if the digit rotates to the position of 3 o'clock or 9 o'clock, then both rewards are $-90$.

### H.3  DETAILS ABOUT CONFOUNDING SIMULATIONS

The probabilities between the confounder, action and reward are presented as follows,

$$p(u = 0) = 0.8, \quad p(u = 1) = 0.2; \tag{24}$$
$$p(a = T_1|u = 0) = 0.24, \quad p(a = T_2|u = 0) = 0.76; \tag{25}$$
$$p(a = T_1|u = 1) = 0.77, \quad p(a = T_2|u = 1) = 0.23; \tag{26}$$
$$p(r_c = R_1|a = T_1, u = 0) = 0.93, \quad p(r_c = R_2|a = T_1, u = 0) = 0.07; \tag{27}$$
$$p(r_c = R_1|a = T_2, u = 0) = 0.87, \quad p(r_c = R_2|a = T_2, u = 0) = 0.13; \tag{28}$$
$$p(r_c = R_1|a = T_1, u = 1) = 0.73, \quad p(r_c = R_2|a = T_1, u = 1) = 0.27; \tag{29}$$
$$p(r_c = R_1|a = T_2, u = 1) = 0.69, \quad p(r_c = R_2|a = T_2, u = 1) = 0.31; \tag{30}$$
$$\mu_{r_c=R_1} = -1, \quad \mu_{r_c=R_2} = -200, \quad \sigma = 2. \tag{31}$$

Note that, in our model $r_c$ is a function of $u$, $a_t$, and $z_t$. We omit the notation of the state in the conditional probability of $r_c$ because $z_t$ is fixed at each time step, which does not affect the probability. To help readers understand this setting, we can make a straightforward analogy. More specifically, the confounder $u$ stands for the socio-economic status where $u = 0$ represents the poor and $u = 1$ the rich. Treatment $T_1$ is more expensive but more valid than Treatment $T_2$. $R_1$ means a healthier feedback/recovery than $R_2$. Equation (24) says that the poor are much more than the rich (it is indeed in reality). Equation (25) and Equation (26) tell us the fact that the poor tend to choose the cheaper but less valid treatment $T_2$ whilst the rich prefer the more expensive but more valid treatment $T_1$, which also makes sense in the real world. Finally, Equations (27)-(30) reveal that $T_1$ is a better treatment than $T_2$ within both poor and rich population.

## I  ADDITIONAL EXPERIMENTAL RESULTS

As shown in Figure 7 and Figure 8.

## J  EXPERIMENTAL SETTINGS

As shown in Table 1 and Table 2.

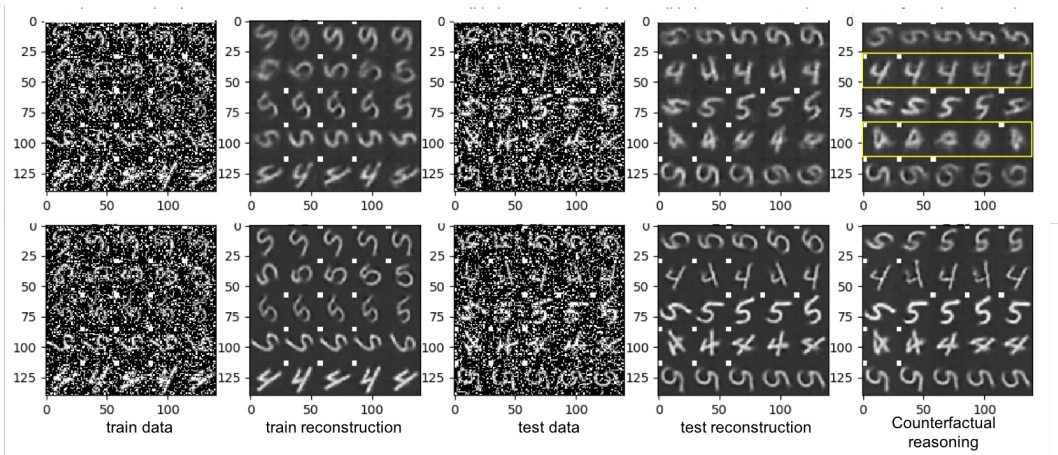

Figure 7: Reconstruction and counterfactual reasoning on the confounding MNIST dataset. Top row: results from the model without the confounder $u$ ($M_{\text{orin}}$); Bottom row: results from the model with the confounder $u$ ($M_{\text{decon}}$). The sequences boxed in yellow have non-consecutive squares.

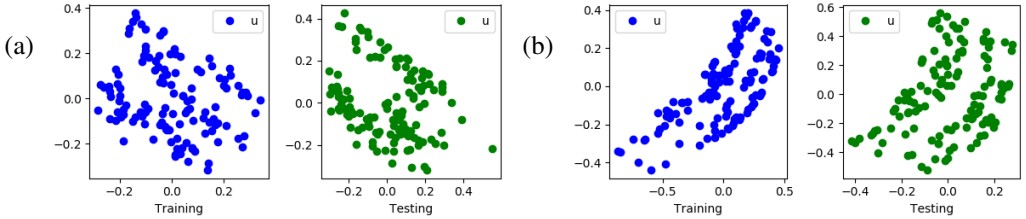

Figure 8: (a) Plot of 128 data points sampling from the posterior approximate of $u$ on the confounding MNIST dataset; (b) Plot of 128 data points sampling from the posterior approximate of $u$ on the confounding CartPole dataset;

| Hyperparameter | Value |
|---|---|
| **Deconfounding Model** | |
| learning rate | 0.0001 |
| dimension of $z_t$ | 50 |
| dimension of $x_t$ | 784 |
| dimension of $a_t$ | 1 |
| dimension of $r_t$ | 1 |
| dimension of $u$ | 2 |
| dimension of LSTM unit | 100 |
| batch size | 128 |
| number of steps | 5 |
| number of epoch | 400 |
| **Deconfounding AC** | |
| number of episodes | 1500 |
| number of time steps in each episode | 200 |
| capacity of the replay memory | 100,000 |
| batch size | 128 |
| sample size of $u$ | 200 |

Table 1: Hyperparameters for deconfounding reinforcement learning. Note that, we set the dimension $u$ to 1 in Section 4.4 for simplicity.

| Function | Architecture |
|---|---|
| | **Deconfounding Model** |
| $f_1, f_2$ | FC$_{512}$ → Conv$_{7,32}$ → Conv$_{14,16}$ → Conv$_{28,1}$ → FC$_{784}$ → {sigmoid, softplus} |
| $f_3, f_4$ | {FC$_{100}$, FC$_{100}$} → FC$_{200}$ → FC$_{200}$ → FC$_1$ → {tanh, softplus} |
| $f_5, f_6$ | {FC$_{100}$, FC$_{100}$, FC$_{100}$} → FC$_{100}$ → FC$_{100}$ → FC$_1$ → {sigmoid, softplus} |
| $f_7, f_8$ | {FC$_{100}$; FC$_{100}$} → FC$_{100}$ → FC$_{100}$ → FC$_{50}$ → {None, softplus} |
| $f_9, f_{10}$ | {[Conv$_{5,16}$ → Conv$_{5,32}$ → FC$_{100}$], FC$_{100}$, FC$_{100}$} → FC$_{100}$ → FC$_{100}$ → LSTM$_{100,5}$ → FC$_1$ → {None, softplus} |
| $f_{11}, f_{12}$ | {[Conv$_{5,16}$ → Conv$_{5,32}$ → FC$_{100}$], FC$_{100}$, FC$_{100}$} → FC$_{100}$ → FC$_{100}$ → LSTM$_{100,5}$ → FC$_{50}$ → {None, softplus} |
| $f_{13}, f_{14}$ | Conv$_{5,16}$ → Conv$_{5,32}$ → Conv$_{5,32}$ → FC$_1$ → {tanh, softplus} |
| $f_{15}, f_{16}$ | {[Conv$_{5,16}$ → Conv$_{5,32}$ → Conv$_{5,32}$ → FC$_{100}$], FC$_{100}$} → FC$_{100}$ → FC$_1$ → {sigmoid, softplus} |
| | **Deconfounding AC** |
| $V(z_t; \phi_V)$ | FC$_{300}$ → FC$_{300}$ → {None, softplus} |
| $\pi(a_t|z_t; \theta)$ | FC$_{300}$ → FC$_{300}$ → {tanh, softplus} |

Table 2: Architectures for deconfounding reinforcement learning. Here FC$_k$ stands for a fully-connected layer with $k$ units, Conv$_{k,n}$ for a convolution layer with $n$ filters of size $k \times k$, LSTM$_{n,t}$ for a LSTM layer rolling out for $t$ steps with latent size of $n$, $\{\cdot\}$ for the parallel operators, and $[\cdot]$ for the sequential operators. In default, FC$_k$ and Conv$_{k,n}$ are followed by a softplus activation layer and a batch-norm layer, which are omitted here for simplicity. Note that, in our setting, the two functions in each pair $\{f_{2i-1}, f_{2i}\}_{i=1,\ldots,8}$ share the same parameters except for the last layer.

