# OpenReview forum: "Deconfounding Reinforcement Learning in Observational Settings"
_ICLR.cc/2019/Conference_

### Official Review · AnonReviewer1 · 2018-11-02
**Setting doesn't make sense for RL and experiments don't evaluate causal questions**

**Rating:** 2
**Confidence:** 4

**Review:**

This paper presents a method for reinforcement learning (RL) in settings where the relationship between action and reward is confounded by a latent variable (unobserved confounder). While I firmly believe that RL would benefit from taking causality more seriously, this paper has many fatal flaws that make it not ready for publication.

First, and most importantly, the paper is unclear about the problem it is trying to solve. It talks about confounded RL as being settings in which a confounder affects both the action and reward. In typical RL settings this wouldn’t make sense: in RL you get to choose the policy so it doesn’t make sense to assume that the choice of action is confounded while you’re doing RL. To get around this, the authors assume that they’re working with observational data and doing RL on a generative model leant from the observational data. But by doing this, they have assumed away the key advantage that RL has over causal inference: the ability to experiment in the world. The authors justify this assumption by considering high-stakes settings where experimentation is either too risky or too costly, but they don’t explain why you would want to do RL at all when you could just do causal inference directly. If you can’t experiment, RL offers no advantages over standard causal inference methods and bring serious disadvantages (sample-efficiency, computational cost, etc.).

# Method
The authors learn a variational approximation to a particular graphical model that they assume for their RL setting. They then treat the variational approximation as the true distribution which allows them to perform causal inference via the backdoor correction. They claim this is identified but this is false - it is only identified with respect to the variational distribution, not the true distribution and we have no a priori reason to  believe that the variational distribution well-approximated the true distribution. In principle, the authors could have tested how well this works experimentally but their experimental setup has problems which prevent this being evaluated.

Quibbles:
 - Page 3: the authors claim the model is “without loss of generality” but this is false - there are many settings that would not conform to this model: e.g. the multi agent settings that economics studies; health settings with placebo effects where reward depends on observations directly; etc.
  - Page 4 above the equations: either the equations describe the variational approximation to the generative model or the equations shouldn’t all be factorized normal distributions. Real data isn’t made up of factorized normals.

# Experiments

The authors evaluate their method on three simulated datasets: Confounding MNIST, Confounding Cartpole and Confounding Pendulum. All three have the same methodological problems so I’ll only focus on the MNIST dataset. They synthesize their MNIST dataset but corrupting a subset of MNIST digits with noise and treating actions as rotations. Rewards are given by the absolute difference in angle between the rotated digit and the original unrotated digit. “Confounding” is added by having a binary latent variable affect the amount that the digit is rotated - but importantly, the reward isn’t affected directly by the latent variable. Because of this, there isn’t actually a confounding problem - the “confounder” simply changes the rotation of the digit and can be treated as additional experimentation from the perspective of causal inference. The authors evaluate their method by examining reconstructions of the MNIST digit, but this simply checks how well the variational inference is working, not whether the causal inference is working (there would be no way to evaluate the latter on this dataset because there is no confounding). Effectively all they find is a better-designed variational distribution will do a better job of reconstructing the input (without modelling the latent u, the VAE is forced to average over its two states resulting in more blurry samples).

The RL evaluations aren’t described in enough detail to conclusively explain the difference observed, but it seems to be driven by the fact that the standard RL methods are working with worse variational approximation distributions.

# Summary
This work studies a setting in which the correct baselines would be causal inference algorithms (but they aren’t considered) and their experimental evaluation has serious flaws that prevent it supporting the claims made in the paper.

---

> ### Author Response · Authors · 2018-11-25
> **Rebuttal**
>
> Thanks for your comments. We have updated the paper and made the paper much clearer. We recommend you re-read the full updated paper to get some refreshed stuff.
>
> Re motivation: Refer to Section 1.
>
> Re the method: Refer to Section 2.4, Section 3.3, and Appendix A.
>
> Re quibbles: Refer to Section 3.1 and Section 3.4. We have to emphasize that the factorization assumption in terms of Gaussian is a widely used technique in machine learning communities. I agree that the real data is not made up of factorized Gaussians, but the assumption of factorized Gaussian is the first step and also the easiest way to deeply understand how the proposed approach works. It is not necessary to get ourselves lost in the complicated distributions, which is not beneficial to capturing some insights about the nature of the model. More importantly, even with such simplified assumption, experiments presented in Section 4.3 and Section 4.4 show that our model work much better. Especially as shown in Figure 3(c), in each episode our deconfounding algorithm almost chooses the optimal action at each time step, whilst the vanilla algorithm makes a wrong decision for more than half time.
>
> Re experiments: Refer to the whole Section 4 and the corresponding appendices. Note that, in this new draft we include all the details about the experiments, especially about how to design a reasonable reward and Appendix H.3 also provides a straightforward analogy to help readers understand the design. As mentioned above, Figure 3(c) demonstrated that the confounder u plays an extremely important role in reinforcement learning algorithms, because in that experiment, in each episode our deconfounding RL algorithm almost chooses the optimal action at each time step, whilst the vanilla RL algorithm makes a wrong decision for more than half time.

---

> ### Author Response · Authors · 2018-11-26
> **More Detailed Rebuttal**
>
> Thanks for your comments. We have updated the paper and made the paper much clearer. We recommend you re-read the full updated paper to get some refreshed stuff.
>
> Re motivation and identification: Refer to Section 1.
>  In this paper, we propose a general formulation to cope with a family of reinforcement learning tasks in observational settings, that is, learning good policies solely from the historical data produced by real environments with confounders (i.e., the factors affecting both actions and rewards). Actually, in recent years, reinforcement learning (RL) has made great progress, spawning a large number of successful applications especially in terms of games (Silver et al., 2016; Mnih et al., 2013; Ope- nAI, 2018). Within this background, much attention has been devoted to the development of RL algorithms with the goal of improving treatment policies in healthcare (Gottesman et al., 2018). In fact, various RL algorithms have been proposed to infer better decision-making strategies for me- chanical ventilation (Prasad et al., 2017), sepsis management (Raghu et al., 2017a;b), and treatment of schizophrenia (Shortreed et al., 2011). In healthcare, a common practice is to focus on the ob- servational setting, because ones do not wish to experiment with patients’ lives without evidence that the proposed treatment strategy is better than the current practice (Gottesman et al., 2018) 1. As pointed out in (Raghu et al., 2017a), even if in the observational setting, RL also has advantages over other machine learning algorithms especially in two situations: when the ground truth of a “good” treatment strategy is unclear in medical literature (Marik, 2015), and when training examples do not represent optimal behavior. On the other hand, although causal inference (Pearl, 2009) has been ex- tensively explored and used in healthcare and medicine (Liu et al.; Soleimani et al., 2017; Schulam & Saria, 2017; Alaa et al., 2017; Alaa & van der Schaar, 2018; Atan et al., 2016), the efficient ap- proach to dealing with time-varying data is still unclear (Peters et al., 2017; Hernn & Robins, 2018). On the basis of the discussion above, in this paper we attempt to combine advantages on both sides to cope with an important family of RL problems in the observational setting, that is, learning good policies solely from the historical data produced by real environments with confounding bias.
>
> Re the method: Refer to Section 2.4, Section 3.3, and Appendix A. The main idea is that we used proxy variables to help identify causal effects of our model (Section 2.4 and Section 3.3). Besides, the causal parameters of our deconfounding model can be identified in the existence of multiple observed proxy variables (Appendix A).
>
>
> Re quibbles: Refer to Section 3.1 and Section 3.4. We have to emphasize that the factorization assumption in terms of Gaussian is a widely used technique in machine learning communities. I agree that the real data is not made up of factorized Gaussians, but the assumption of factorized Gaussian is the first step and also the easiest way to deeply understand how the proposed approach works. It is not necessary to get ourselves lost in the complicated distributions, which is not beneficial to capturing some insights about the nature of the model. More importantly, even with such simplified assumption, experiments presented in Section 4.3 and Section 4.4 show that our model work much better. Especially as shown in Figure 3(c), in each episode our deconfounding algorithm almost chooses the optimal action at each time step, whilst the vanilla algorithm makes a wrong decision for more than half time.
>
> Re experiments: Refer to the whole Section 4 and the corresponding appendices. Note that, in this new draft we include all the details about the experiments, especially about how to design a reasonable reward and Appendix H.3 also provides a straightforward analogy to help readers understand the design. As mentioned above, Figure 3(c) demonstrated that the confounder u plays an extremely important role in reinforcement learning algorithms, because in that experiment, in each episode our deconfounding RL algorithm almost chooses the optimal action at each time step, whilst the vanilla RL algorithm makes a wrong decision for more than half time.

---

### Official Review · AnonReviewer2 · 2018-11-03
**Strong and important idea - presentation and execution can be improved**

**Rating:** 4
**Confidence:** 4

**Review:**

The paper addresses an important and often overlooked issue in off-policy reinforcement learning - the possibility of confounding between the agent's actions and the rewards. This is a subject which has been exhaustively explored in the causal inference literature, and the authors are very correct in suggesting that it should be incorporated into the world of reinforcement learning.  Specifically they propose a generative model with a global latent confounder that is inferred using a variational autoencoder architecture.

The paper is generally well-written, though some points could be made clearer in my opinion, as detailed below. The experiments are constructed by introducing confounding into existing datasets; performance seems to be good, but I am not entirely sure whether the given architecture is necessary, see comments below.

High-level comments:
(1) Classic RL deals with confounders all the time. The state is a confounder between the action and the reward. The issue of confounding becomes less trivial when one is performing off-policy RL when the original policy is *unknown*. This is exactly the case that the authors mention when they cite the recent work by Gottesman et al. (2018) who deal with using RL to learn from the actions of physicians in a hospital.  While I am sure the authors are aware of these distinctions, I think the paper would be better if this is spelled out very explicitly. This includes explaining why this issue doesn't come up in classic RL.

(2) Assuming the case above - off-policy RL with unknown confounders - one would usually assume "no unmeasured confounding", i.e. that the observed actions are an unknown but learnable function of the observed states. That is basically the scenario of most off-policy RL.

(3) However, the authors strive to go one step beyond the case (2), to a situation where there is an *unmeasured* confounder affecting both observed actions and rewards. If nothing is known about this unmeasured confounder, then it is generally impossible to learn effective policies, as the causal effects of actions are not identifiable from the observed data. In this paper, the authors make an implicit assumption that while the confounder is unmeasured, it can still be inferred from the data. This is an intermediate step between "no unmeasured confounding" and "complete unmeasured confounding". This is related to work on using proxy variables e.g. Kuroki & Pearl (2014) and even more closely related to the work cited by Louizos et al. (2017).
Again, I think the paper would be much improved if all this is addressed explicitly.

(4) An important consequence of point (3) above is that in fact adding the single global latent-confounder U is not, in itself, very important from a causal perspective. The sequence of variables Z_1... Z_T are already latent confounders that are assumed to be inferrable from data. It is true that the addition of the global U might change the statistical and optimization properties of the model. This leads to a very important conclusion: the authors should test their model with and without U. I think this specific ablation experiment is crucial. In many cases I am sure that the assumption of a global latent confounder is a good one and is especially useful in the VAE case where it will make optimization more stable. However, in principle, all of U's roles could be taken within the sequence of Z's, and I am curious to see in practice how big of an effect it has.

(5) I wish to add that even if the U variable turns out to not add much empirically, this work is still valid since the sequence of Z's can themselves be considered inferred latent confounders.

Specific comments:
(1) 2.3: there are more than 2 ways of computing the do-operator. RCTs and backdoor are the best known approaches, but not the only ones, e.g. there is frontdoor adjustment.

(2) I think the paper would be easier to follow if there was one concrete example used throughout. This will make it easier to understand and possibly verify/criticize the assumptions of the generative model.

(3) Related to "higher-level point (4)" above, in eqs. 17 & 18 note that Z_t is unknown, same as U. Both are inferred. This also leads to the question which Z_t is actually used in practice? Is it the mean, or is it also sampled from the approximate posterior q?

(4) Below eq. 19, it would be very useful for the readers if you could explain exactly when would there be a difference between the two versions p(r_{t+1}|z_t,a_t) and p(r_{t+1}|z_t, do(a_t=a))

(5) In the description of all the experiments I was missing a crucial point: how does the introduced confounder affect the reward? Is it only through the different actions? The way it is currently explained, it seems like the added variable introduces lack of *overlap*, but not strictly confounding.

(6) The description of the experiment in 4.3 could be more detailed. What exactly was the training and test? What RL method was used? What did the baseline optimize for? I would like to see an ablation experiment where U is not included in the model.

(7) In 4.5, what is the "vanilla" method? And as mentioned above, I would like to see an ablation experiment where U is not included in the model.

---

> ### Author Response · Authors · 2018-11-25
> **Rebuttal**
>
> Thanks for your comment. We have updated the paper and solved all the issues you are concerned about. In the following, we will point out which part in our new draft answers your question, respectively. Nevertheless, we still recommend you re-read the full updated paper to get some refreshed stuff.
>
> Regarding High-level Comments:
>
> Re (1): Refer to Abstract, Section 1, and especially to the last paragraph of Appendix A in which we explained the reason why we consider only adjusting for the confounder u.
>
> Re (2)-(3): Refer to Section 2.4, Section 3.3, and Appendix A for the solution to identification.
>
> Re (4-5): Refer to the last paragraph of Appendix A for the difference between Z and u, and to Section 4.3 and Section 4.4 for both experiments with and without u. It is worth noting that, as shown in Figure 3(c), in each episode our deconfounding algorithm considering u almost chooses the optimal action at each time step, whilst the vanilla algorithm not considering u makes a wrong decision for more than half time.
>
> Regarding Specific Comments:
>
> Re (1): Refer to Section 2.3 and Section 2.4.
>
> Re (2): The kidney stone example is used throughout the paper, referring to Section 1, Section 2.1, Section 2.2, Section 3.1, Footnote 2, Appendix F.2, and Appendix H.3.
>
> Re (3): Z, sampled using Equation (6), has to be used in reinforcement learning algorithms, because we need the state transition when generating trajectories/rollout. Refer to Section 4.1, Section 4.4, and Appendix E.
>
> Re (4): Refer to Appendix F.2 for an intuition of the difference, and to Section 4.4 in which, as shown in Figure 3(c), in each episode our deconfounding algorithm using p(r_{t+1}|z_t, do(a_t=a)) almost chooses the optimal action at each time step, whilst the vanilla algorithm using p(r_{t+1}|z_t,a_t) makes a wrong decision for more than half time.
> Re (5): Refer to Section 4.2 and Appendix H.3 in which a straightforward analogy is provided as well.
> Re (6): Refer to Section 4.3.
> Re (7): Refer to Section 4.4.

---

> ### Author Response · Authors · 2018-11-26
> **More Detailed Rebuttal about High-level Comments**
>
> Regarding High-level Comments:
>
> Re (1): Refer to Abstract, Section 1, and especially to the last paragraph of Appendix A in which we explained the reason why we consider only adjusting for the confounder u.
> Note that, in our model we have to differentiate the two types of confounders: the time-independent confounder u and time-dependent confounders {z_t}(t=1,...,T), each playing a respective role in the model. The former, as a global confounder, will affect the whole course of treatment, and therefore should be adjusted for. In the example of kidney stones, the existence of the confounder (i.e., the size of stones) will lead to a wrong treatment if not adjusting for it. In contrast, the time-varying confounders {z_t} act as states in RL, which, in principle, should not be adjusted for, because the goal in RL is to learn a good policy in which any action is indeed supposed to be conditional on a specific state. On the other hand, in terms of rewards, what an agent expects at each time step is exactly the immediate reward when taking a specific action at a specific state, without the need of adjusting for states. It is worth noting that the case with time-varying confounders {z_t} can be thought of to meet a pseudo or weak causal sufficiency assumption under which the causal effects of actions on rewards will not be influenced by states at each time step (Zhang et al., 2017; 2015). This key difference motivates us to only adjust for the time-independent confounder u in this paper. In addition, as shown in Figure 1(b), in our case where a policy depending only on the confounder is applied to generating the data (Section 4.2), the arrow from z_t to at is not necessary so that z_t can be viewed as not a confounder of at and r_{t+1}. This also provides another reason why we do not need to adjust for z_t.
>
> Re (2)-(3): Refer to Section 2.4, Section 3.3, and Appendix A for the solution to identification. The main idea is that we used proxy variables to help identify causal effects of our model (Section 2.4 and Section 3.3). Besides, the causal parameters of our deconfounding model can be identified in the existence of multiple observed proxy variables (Appendix A).
>
> Re (4-5): Refer to the last paragraph of Appendix A for the difference between Z and u, and to Section 4.3 and Section 4.4 for both experiments with and without u. It is worth noting that, as shown in Figure 3(c), in each episode our deconfounding algorithm considering u almost chooses the optimal action at each time step, whilst the vanilla algorithm not considering u makes a wrong decision for more than half time.

---

> ### Author Response · Authors · 2018-11-26
> **More Detailed Rebuttal about Specific Comments**
>
> Re (1): Refer to Section 2.3 and Section 2.4, where we describe more methods of adjusting for confounders.
>
> Re (2): The kidney stone example is used throughout the paper, referring to Section 1, Section 2.1, Section 2.2, Section 3.1, Footnote 2, Appendix F.2, and Appendix H.3.
>
> Re (3): Z, sampled using Equation (6), has to be used in reinforcement learning algorithms, because we need the state transition when generating trajectories/rollout. Refer to Section 4.1, Section 4.4, and Appendix E.
>
> Re (4): Refer to Appendix F.2 for an intuition of the difference, and to Section 4.4 in which, as shown in Figure 3(c), in each episode our deconfounding algorithm using p(r_{t+1}|z_t, do(a_t=a)) almost chooses the optimal action at each time step, whilst the vanilla algorithm using p(r_{t+1}|z_t, a_t) makes a wrong decision for more than half time.
>
> Re (5): Refer to Section 4.2 and Appendix H.3 for the details about how to define the confounding datasets in which the reward exactly depends on the action and the confounder. Also, a straightforward analogy of kidney stones to this confounding dataset is provided in Appendix H.3 as well.
>
> Re (6): Refer to Section 4.3.
> Actually, to demonstrate the validity of our deconfounding model, denoted by M_decon, we compare with the original model (i.e., the model similar to that shown in Figure 1(b) but without the confounder u), denoted by M_orin. We train M_decon by optimizing Equation (19) but train Morin by a little different loss function excluding the confounder u whose full derivation can be found in Appendix C. Both models are separately trained in a batch manner on the training set (i.e., 140K sequences of length five of images) of the confounding dataset. Afterwards, following the steps depicted in Section 4.1, we use each trained model to perform the reconstruction task on the training set, and both reconstruction and counterfactual reasoning tasks on the testing set (i.e., 28K sequences of length five of images). Figure 2 presents a comparison of M_decon and M_orin, in terms of reconstruction and counterfactual reasoning on the confounding Pendulum dataset. The second row is based on M_decon (Figure 1(b)), whilst the top row comes from Morin. It is evident that the results generated by the deconfounding model is superior to those produced by the model not taking into account the confounder. To be more specific, as shown in the zoom of samples on the bottom row, Morin generates more blurry images than M_decon, because, without modelling the confounder u, M_orin is forced to average over its multiple latent states resulting in more blurry samples.
>
> Re (7): Refer to Section 4.4.
> we will evaluate the proposed deconfounding actor-critic (AC) method by comparing with its vanilla version on the confounding Pendulum dataset. In the vanilla AC method, given a learned M_orin, we optimize the policy by calculating the gradient presented in Equation (21) on the basis of the trajectories/rollouts generated through M_orin. Equation (21) involves two functions: V (z_t; φ_V ) and π(a_t|z_t; θ), whose parameters can be found in Appendix J. It is worth noting that, in this vanilla case, each reward r_{t+1} is produced from the conditional distribution p(r_{t+1}|z_t, a_t). In contrast, the proposed deconfounding AC method is built on M_decon. Although the same gradient method (Equation (21)) is utilized to optimize the policy, we base the deconfounding AC approach on the different trajectories/rollouts generated by M_decon in which each reward r_{t+1} relies on the interventional distribution p(r_{t+1}|z_t, do(a_t)) computed using Equation (20).
>
> In the training phase, for both vanilla AC and deconfounding AC, we run a respective experiment over 1500 episodes with 200 time steps each. In order to reduce non-stationarity and to decorrelate updates, the generated data is stored in an experience replay memory and then randomly sampled in a batch manner (Mnih et al., 2013; Riedmiller, 2005; Schulman et al., 2015; Van Hasselt et al., 2016). In each episode, we summarize all the rewards and further average the sums over a window of 100 episodes to obtain a smoother curve. As shown in Figure 3(a), obviously our deconfounding AC algorithm performs significantly better than the vanilla AC algorithm in the confounded environment.
>
> In the testing phase, we first randomly select 100 samples from the testing set, each starting a new episode, and then use the learned policies to perform reasoning over 200 time steps as we did during the training time. From the resulting 100 episodes, we plot the total reward for each, shown in Figure 3(b), and compute the percentage of the optimal action T1 in each episode, presented in Figure 3(c). It is worth noting that Figure 3(c) tells us that in each episode our deconfounding AC almost chooses the optimal action at each time step, whilst the vanilla AC makes a wrong decision for more than half time.

---

### Official Review · AnonReviewer3 · 2018-11-03
**interesting problem**

**Rating:** 4
**Confidence:** 3

**Review:**

I have read the discussion from the authors. my evaluation stays the same.
--------
this paper studies an interesting question of how to learn causal effects from observational data generated from reinforcement learning. they work with a very challenging setting where an unobserved confounder exists at each time step that affects actions, rewards and the confounder at next time step.

the authors fit latent variables models to the observational data and perform experiments.

the major concern is on the causal inference side, where it is not easy to claim anything causal in such a complicated system with unobserved confounders. causal inference with unobserved confounders cannot be simply solved by fitting a latent variable model. there exists negative examples even in the simplest setting that two distinct causal structure can lead to the same observational distribution. for example here, https://www.alexdamour.com/blog/public/2018/05/18/non-identification-in-latent-confounder-models/

it could be helpful if the authors can lay out the identification assumptions for causal effects. before claiming anything causal and justifying experimental results.

---

> ### Author Response · Authors · 2018-11-25
> **Rebuttal**
>
> Thanks for your comments. We have updated the paper and please refer to Section 2.4, Section 3.3, and Appendix A of the new version for the solution to identification.

---

> ### Author Response · Authors · 2018-11-26
> **More Detailed Rebuttal**
>
> Thanks for your comments. We have updated the paper and please refer to Section 2.4, Section 3.3, and Appendix A of the new version for the solution to identification. The main idea is that we used proxy variables to help identify causal effects of our model (Section 2.4 and Section 3.3). Besides, the causal parameters of our deconfounding model can be identified in the existence of multiple observed proxy variables (Appendix A).

---

### Public Comment · (anonymous) · 2018-10-13
**Formula Check**

In Eq (10) Page 5, equation written is,
q(z, u | x, a, r) = q(u | x, a, r).q(z | x, a, r)

But q(u | z, x, a, r) != q(u | x, a, r) as v-structure z1 --> r2 <-- u opens up.

This will affect Eq (12), Eq (18) etc. Please have a look.

---

> ### Author Response · Authors · 2018-10-13
> **It is actually the widely used factorization assumption in variational inference.**
>
> Thank you for your interest in our paper and for checking our formula.
>
> In the model part, you are exactly right, where p(z, u | x, a, r) != p(u | x, a , r) p(z | x, a, r). However, in the variational inference part, for simplicity, we use the well known trick, factorization assumption, to obtain q(z, u | x, a, r) = q(u | x, a , r) q(z | x, a, r) as we claimed on Page 13.
>
> Actually, we can think q(u | z, x, a, r) = q(u | x, a, r) because (x, a, r) already contains all the information about z.

---

> > ### Public Comment · (anonymous) · 2018-10-14
> > **Training pipeline**
> >
> > Amazing! I really like the idea.
> >
> > It would have been nice if you had specified training pipeline which tells what is trained after what.
> >
> > Can you elaborate on that?

---

> > > ### Author Response · Authors · 2018-10-14
> > > **Thank you for your useful feedback.**
> > >
> > > Thank you for your useful feedback and we are glad you like the idea.
> > >
> > > As described in the last paragraph on Page 1, generally speaking, the training pipeline consists of two steps:
> > > Step 1: Given the time-independent confounding assumption (Section 3.1), we learn the deconfounding model, as presented in Fig 2, from the observational data;
> > > Step 2: We optimise the policy or Q-function based on the deconfounding model we learned in Step 1.
> > >
> > > More specifically, in Step 1, given the observational data (x, a, r), we optimise the variational lower bound (Eq.11) with two extra terms (Eq.15 and Eq.16). Once the deconfounding model is learned, we know the state transition function p(z_t | z_{t-1}, a_{t-1}) and can also calculate the deconfounding reward function p(r_t | z_t, do(a_t)) according to Eq.17. In Step 2, we treat the learned deconfounding model as a RL environment like CartPole in OpenAI Gym, and directly exploit it to generate trajectories/rollouts through the state transition function and the deconfounding reward function. On the basis of the generated trajectories/rollouts, we can train the Q-network using Eq.19 or the policy network using Eq.20.
> > >
> > > We will clarify this in the new draft. Thank you.

---

> > > > ### Public Comment · (anonymous) · 2018-10-16
> > > > **N value in Eq 18**
> > > >
> > > > What value of N is chosen in Eq 18?
> > > >
> > > > While plotting Fig 5 N=128, but is it the value chosen even for experimental results?

---

> > > > > ### Author Response · Authors · 2018-10-16
> > > > > **Trade-off**
> > > > >
> > > > > As we know, there always is a trade-off between time and accuracy in MC methods. Considering this balance, we set N= 400 for experimental results.

---

> > > ### Public Comment · (anonymous) · 2018-11-03
> > > **Sampling u in Eq 18.**
> > >
> > > Shouldn't u be sampled from the model because the Deconfounding Q learning pipeline starts after the model is built which doesn't have access to observations (x,a,r)?

---

> > > > ### Author Response · Authors · 2018-11-04
> > > > **Response**
> > > >
> > > > Yes, u is sampled from the model. In Equation (18), (x, a, r) in the posterior q(u|x, a, r) are estimated from the model rather than the observations. We will use different notations in the updated version.

---

> > > > > ### Public Comment · (anonymous) · 2018-11-04
> > > > > **Why not just sample from p instead of q?**
> > > > >
> > > > > You have built your model "p". Use it to sample u values. Why use q anyway? Why need to estimate x,a,r from z. That's a convoluted approach.

---

> > > > > > ### Author Response · Authors · 2018-11-04
> > > > > > **Response**
> > > > > >
> > > > > > Re "Why use q anyway?"
> > > > > > Please keep in mind that u is a latent variable. q(u|x, a, r) is the posterior containing the information from the data whilst p(u) is nothing but a prior.
> > > > > >
> > > > > > Re "Why need to estimate x,a,r from z."
> > > > > > Because q(u|x, a, r) depends on (x, a, r) which are unknown during the testing phase.

---

> > > > > > > ### Public Comment · (anonymous) · 2018-11-04
> > > > > > > **KL Divergence b/w q(u|x,a,r) and p(u) takes care of it**
> > > > > > >
> > > > > > > KL loss ensures two distributions are close by. Draw as many "u" values from p(u) instead of estimating x,a,r.

---

> > > > > > > > ### Author Response · Authors · 2018-11-04
> > > > > > > > **Response**
> > > > > > > >
> > > > > > > > In our case, as shown in Fig 5, the posterior of u has an obvious difference from its unit Gaussian prior, even though their KL loss converged. Therefore, sampling from q is better.

---

### Public Comment · (anonymous) · 2018-10-15
**Counterfactual claim is not really "Counterfactual" nor "Interventional"**

Thoughts:
(1)Paper claims to have done counterfactual reasoning which according to Pearl's literature(as cited in paper) is not counterfactual. Counterfactual has to incorporate hindsight -> What action a(t) will be taken if you are in state x(t) "given the fact" that different state x'(t) and action a'(t) were taken. "Given the fact" makes all the difference.

Claim:
Paper generates samples based on conditionals

Edit:
(2)Why not additional experiment on state intervention as q(action=a | state=x) != q(action=a | do(state=x)) where inequality arises due to back door path x <- z -> a opening up. This changes equations accordingly.

---

> ### Author Response · Authors · 2018-10-16
> **It actually is.**
>
> Thank you for your interest in our paper. Happy to discuss with you all the stuff about causal concepts.
>
> Re Thought (1): What you understand about counterfactual reasoning is exactly right, which is also what we did in the paper. Given the fact that we know the training set (e.g., we knew the fact that what happened when we took action a_1 at x_1 in the training set), we want to know “what would have happened had we taken a different action a_2 at an unseen x_2 in the testing set?” That is exactly what counterfactual reasoning does. Note that, like ones did in [1], we primarily emphasise the inference on unseen data, which plays a pivotal role in the following RL part.
>
> Re Thought (2): in our paper, we do not have the interventional distribution p(a | x).
> a) If what you meant is q(a | x), it is an auxiliary distribution in the variational inference part, for which we did not take intervention into account, because that is not what we studied in the paper;
> b) If you meant p(a | z), that is actually the policy function in RL. Usually we also do not factor the intervention into the policy, because the definition of the policy is a conditional distribution.
> Actually we only considered the intervention in the reward function p(r | z, do(a)) where z is treated as a fixed constant at each time step.
>
> We hope this helps.
>
> [1] Rahul G Krishnan, Uri Shalit, and David Sontag. Deep kalman filters. arXiv preprint arXiv:1511.05121, 2015.

---

> > ### Public Comment · (anonymous) · 2018-10-16
> > **Good one!**
> >
> > Given the training data incorporates for hindsight.

---

### Public Comment · (anonymous) · 2018-10-28
**Clarification in Architecture**

Lets take an example network f5 ,f6
{FC 100 , FC 100 , FC 100 } → FC 100 → FC 100 → FC 1 → {sigmoid, softplus}

1. Is the output of FC 100, FC 100, FC 100 concatenated and sent to FC100 followed by FC100 and then FC1? .
2. Are the activation functions applied after each layer or at the output of the final FC (FC1)?
3. The activation functions are denoted by a {.}. By definition, does that mean they are parallel operators? In that context what is Mu and sigma for f5 and f6 respectively?  It will be better if you can explain how the activation functions are applied and if possible show a visual representation to get a more intuitive understanding of the network topology

---

> ### Author Response · Authors · 2018-10-28
> **Response**
>
> Thank you for your comment.
>
> Re 1: Exactly.
>
> Re 2: In default, softplus is applied after each intermediate layer, and the explicitly claimed activation functions are applied to the output of the final layer. Here, take f5 and f6 for example, f5 and f6 share the parameters of the first five FC100 layers, each followed by softplus, but have different parameters in their own final layer, i.e., FC1 has two outputs (the output of f5 and the output of f6) which are respectively followed by sigmoid and softplus.
>
> Re 3: In our architecture, each function modelled with a neural network has only one output: mu or sigma^2. Since each pair of functions (e.g., f1 and f2, f5 and f6, etc.) share the parameters of all layers except the final layer, that is, they have their own final layers which are parallel outputs, each followed by a respective activation function. In the case of f5 and f6, as described in Re 2, FC1 has two outputs, each followed by sigmoid and softplus, representing mu (the output of f5) and sigma^2 (the output of f6), respectively. We will clarify this with a figure in the appendix of the new draft. Thank you for your suggestion.

---

> > ### Public Comment · (anonymous) · 2018-10-29
> > **FC1 of f1 and Dimension of u**
> >
> > Clarifications:
> > 1. How is u value chosen? Pg 14
> > 2. How is f1 resized after FC1(512)? Depth not mentioned. Pg 15

---

> > > ### Author Response · Authors · 2018-10-29
> > > **Just as described in the paper**
> > >
> > > Dear reader,
> > >
> > > Re 1: Actually dimension of u can be any number. In the paper we let u=2 because it is easy to visualise it in the 2D plot as shown in Figure 5.
> > >
> > > Re 2: 512 = 4 x 4 x 32, so it can be reshaped to a square 4 x 4 with 32 channels.
> > >
> > > We hope this helps.

---

> > > > ### Public Comment · (anonymous) · 2018-11-02
> > > > **Architecture for measuring q(u|x,a,r) is unclear**
> > > >
> > > > 1>Since u is time independent which particular mean and variance are used to calculate the q(u|x,a,r).
> > > >
> > > > 2> Just to confirm : should q(a|x) and q(r|x,a) be added to the loss function.

---

> > > > > ### Author Response · Authors · 2018-11-02
> > > > > **Response**
> > > > >
> > > > > Re 1. As described in Section 3.3.2, q(u|x, a, r) is modelled in the same way as q(z |x, a, r), each of which is parameterised by a bi-directional LSTM. However, unlike q(z |x, a, r) in which z is time-dependent (i.e., z_t corresponds to x_t, a_t, r_t at each time step), in q(u|x, a, r) u is independent of time steps, meaning that u combines all the information of the whole sequence.
> > > > >
> > > > > Re 2. Yes.

---

> > > > > > ### Public Comment · (anonymous) · 2018-11-03
> > > > > > **Feature independence assumption in p(x|z)?**
> > > > > >
> > > > > > While calculating p(x|z) are you assuming that all the features (784) are independent? As X is 784 dimensional, calculating p(x|z) will give numerical instability. How are you mitigating this?

---

> > > > > > > ### Author Response · Authors · 2018-11-04
> > > > > > > **Response**
> > > > > > >
> > > > > > > You can think so, because all Gaussians in the paper are assumed to be with diagonal covariance matrices.

---

### Comment · AnonReviewer1 · 2018-10-31
**Clarification on confounding MNIST**

1. What is the purpose of the "sequence of three consecutive squares (2 x 2 in pixel)"? Are they added before or after the rotation?
2. What is the "some policy" that is perform on the images? Random rotations? Or rotations toward vertical?
3. Does the confounder, u, only affect the magnitude of rotation? i.e. rotation given u = 1 is between 22.5 and 45, while u=0 it is between 0 and 22.5? As far as I can see, u doesn't affect the reward directly? Is that correct?

---

> ### Author Response · Authors · 2018-10-31
> **Response**
>
> Thank you for your comment.
>
> Re 1. We exactly followed the same setting of Healing MNIST in [1]. As [1] said, “the squares within the sequences are intended to be analogous to seasonal flu or other ailments that a patient could exhibit that are independent of the actions and which last several timesteps”. We want to show that our model can learn long-range patterns, which plays an important role in medical applications. The squares are added after the rotation.
>
> Re 2. “Some policy” can be any policy, e.g., random rotations or rotations toward vertical. But in our case, considering the confounder, we used the policy where action is affected by the confounder u.
>
> Re 3. Simply speaking, the confounder not only affects the action through the magnitude, 22.5 ≤ |a| ≤ 45 or 0 ≤ |a| < 22.5, but also affects the reward through the direction (i.e., clockwise or counterclockwise), a or -a, where a and -a will result in different rewards.
>
> We will clarify these in the new draft. Thank you for your suggestion.
>
> [1] Rahul G Krishnan, Uri Shalit, and David Sontag. Deep kalman filters. arXiv preprint arXiv:1511.05121, 2015.

---

> > ### Public Comment · (anonymous) · 2018-11-01
> > **MNIST Reconstruction without Confounder**
> >
> > To get reconstructed image in Fig 3 top row, are you training a separate architecture over changed loss without u? Or is there a clever hack to mitigate retraining?

---

> > > ### Author Response · Authors · 2018-11-01
> > > **Response**
> > >
> > > Exactly, we trained a separate architecture without u.

---

### Public Comment · (anonymous) · 2018-11-19
**NaN issue during training**

Pros:
We were thinking of a reward graph with and without "u" to see how it will make a difference during variational inference step but actor-critic algorithms show that no matter what difference it makes during training the model, it sure has an impact while learning in actor critic. (+1 for paper update)

Issues:
While reproducing experiments, we are getting NaN loss during training. We have clipped gradients to -1 to 1, Xavier initialized layers, Z-normalized actions/rewards, L2 regularized dense layers and used same architecture. Any additional things to be taken care of?

---

> ### Author Response · Authors · 2018-11-26
> **Batch-Norm**
>
> Batch Norm is necessay, see the last page of the updated paper.

---

### Author Response · Authors · 2018-11-25
**To all reviewers**

Thanks to all the reviewers. In the updated paper, we have addressed all the issues the reviewers are concerned about. If you have more questions, please feel free to contact us.

---

### Meta-Review · Area_Chair1 · 2018-12-14
**Interesting work, with unclear motivation and relation to previous work**

**Confidence:** 4
**Recommendation:** Reject

**Metareview:**

The paper studies RL based on data with confounders, where the confounders can affect both rewards and actions.  The setting is relevant in many problems and can have much potential.  This work is an interesting and useful attempt.  However, reviewers raised many questions regarding the problem setup and its comparison to related areas like causal inference.  While the author response provided further helpful details, the questions remained among the reviewers.  Therefore, the paper is not recommended for acceptance in its current stage; more work is needed to better motivate the setting and clarify its relation to other areas.

Furthermore, the paper should probably discuss its relation to (1) partially observable MDP; and (2) off-policy RL.